# Characterizing the distributions of IDO-1 expressing macrophages/microglia in human and murine brains and evaluating the immunological and physiological roles of IDO-1 in RAW264.7/BV-2 cells

Rong Ji[1]☉, Lixiang Ma[2]☉, Xinyu Chen[2], Renqiang Sun[3], Li Zhang[1]*, Hexige Saiyin🄳[4]*, Wenshi Wei[1]*

1 Department of Neurology, Huadong Hospital, Fudan University, Shanghai, China, 2 Department of Anatomy, Histology & Embryology, School of Medical Sciences, Shanghai, China, 3 IBS, Fudan University, Shanghai, China, 4 State Key Laboratory of Genetic Engineering, School of Life Sciences, Fudan University, Shanghai, P.R. China

☉ These authors contributed equally to this work.
* saiyin@fudan.edu.cn (HS); wenshiwei1999@163.com (WW); lizhang_huadong@163.com (LZ)

## Abstract

Indoleamine 2,3-dioxygenase 1 (IDO-1) is an immunosuppressive enzyme expressed in the placenta, neoplastic cells, and macrophages to reject T cells by converting tryptophan into kynurenine. However, the role of IDO-1 in brain immunity, especially in the meninges, is unclear. We aim to elucidate the distribution pattern of IDO-1+ macrophages/microglia in the human brain tissues, human glioblastoma, APP/PS1 mouse brains, and quinolinic acid model brains and explore the physiological and immunological roles of IDO-1+ macrophages/microglia. Here, we find that both human and mouse macrophages/microglia of the perivascular and subarachnoid space and in glioblastoma (GBM) expressed IDO-1 but not macrophages/microglia of parenchyma. Using IDO-1 inhibitors including 1-MT and INCB24360, we observed that inhibiting IDO-1 reduced the cellular size and filopodia growth, fluid uptake, and the macropinocytic and phagocytic abilities of human blood monocytes and RAW264.7/BV-2 cells. Inhibiting IDO-1 with 1-MT or INCB24360 increased IL-1β secretion and suppressed NLRP3 expression in RAW264.7/BV-2 cells. Our data collectively show that IDO-1 expression in perivascular and meninges macrophages/microglia increases cellular phagocytic capacity and might suppress overactivation of inflammatory reaction.

## Introduction

Indoleamine 2,3-dioxygenase 1 (IDO-1), an immunosuppressive metabolic enzyme, prevents maternal T cell-driven immune rejection during pregnancy by metabolizing tryptophan into kynurenine [1]. IDO-1 in neoplastic cells, macrophages, and dendritic cells in neoplasia

**Data Availability Statement:** All relevant data are within the paper and its Supporting information files.

**Funding:** Funding, this study was supported by funds from National Natural Science Foundation of China (81371220). Dr. Wenshi Wei received this funding and played a supervision and resources supporter role in this work.

**Competing interests:** The authors declare no competing interests.

suppress T-cell proliferation and natural killer cells, promotes regulatory T-cell (Treg) and myeloid-derived suppressor cell (MDSC) development via tryptophan depletion and kynurenine production, and prevents the overactivation of the immune response [2–4]. Meninges, which cover brain parenchyma, are populated by immune sentinels, including macrophages and dendritic cells, mast cells, T cells, and B cells [5–7]. However, the brain parenchyma rejects T-cell infiltrates [8]. The barriers formed by the pia matter and glia limitans are thought to contribute to T-cell rejection. Whether immune cells in the meninges are involved in T-cell rejection by the parenchyma is unknown.

Although debated for many years, activated macrophages/microglia are classified into the M1 or M2 phenotype for functional annotation [9]. The M1 macrophage is proinflammatory, while the M2 macrophage is anti-inflammatory [9–11]. M1 macrophages increase iNOS and secrete proinflammatory factors, such as TNF-α, IL-1β, IL-6, superoxide, nitric oxide, reactive oxygen species, and proteases [11, 12]. IDO-1 is reported to be highly expressed in M1 macrophages and drives macrophages to express M2 markers such as IL-10 and CXCR4 [13]. IL-4, IL-13, and IL-10 are induced to form typical M2 macrophages [12]. M2 macrophages engulf cellular debris or misfolded proteins, facilitate extracellular matrix (ECM) reconstruction and tissue repair, and support neuronal survival by secreting neurotrophic factors [14, 15]. In viral infections, inhibiting IDO-1 in macrophages leads to a surge in the secretion of proinflammatory cytokines such as IFN-γ, IL-1β, IL-6, and TNF-α [16]. IDO-1$^{-/-}$ mice also secrete type I IFN in LP-BM5 infection and restrict viral replication [17]. If the brain parenchyma utilizes IDO-1 to form an immune barrier and reject T-cell infiltrates, it has not yet been reported. The microglia database (http://research-pub.gene.com/BrainMyeloidLandscape) has shown that in Alzheimer's disease (AD) and Huntington's disease (HD) patients and murine neurodegenerative models (APP/PS1 and PS2/APP), microglia do not upregulate IDO-1, but the microglia in glioblastoma, murine ischemia, cuprizone and LPS models exhibited upregulated IDO-1 [18]. IDO-1 expression in microglia does not depend on aging in either healthy murine models or humans. Isolated human monocytes from the brain expressed a higher level of IDO-1 compared to microglia. Nonparenchymal macrophages in the perivascular space, subdural meninges, and the choroid plexus partially or mainly originate from monocytes [18]. These data provided us with the idea that IDO-1+ might preferentially reside in perivascular space, meninges, and the choroid plexus.

In this work, we detected IDO-1 expression in murine neurodegenerative models and human brain tissue macrophage/microglia and inhibited IDO-1 activities with 1-MT and INCB24360 from testing the roles that IDO-1 plays in macrophage/microglia physiological and immunological activities. Here, we show that IDO-1 is exclusively expressed in macrophage/microglia of the perivascular space, subarachnoid space, and glioblastoma but not in the parenchymal microglia of the brain in humans or murine models. Inhibiting IDO-1 with 1-MT or INCB24360 significantly affects physiological and immunological behavior but not the migration or proliferation of macrophages/microglia, indicating that IDO-1-expressing macrophages in the brain are distinct with the intensive phagocytic ability and lower proinflammatory activity.

## Materials and methods

### Ethics statements

The study's human sample ethics were approved by the Human Ethics Committee of Huashan Hospital at Fudan University (Approval Series Number, 2018–310). Dr. Li Li from Huashan Hospital collected the freshly surged brain samples from GBM patients after a pathologist check, fixed them by 4% PFA, and sectioned them for immunostaining. All mice were cultured

in the animal culturing conditions with air conditioning at the School of Basic Medical Science of Fudan University Animal Cores. All animal procedures were reviewed and approved by the School of Basic Medical Science of Fudan University and Use Committee (Approval Series Number, 2020-0306-002). All mice freely access enough food and sterilized water. A maximum of 5–6 mice was maintained per cage. The infection or injury after surgery were monitored by the veterinarians at the animal cores daily. Mice were deeply anesthetized with isoflurane and perfused by 4% PFA for collecting the brain for immunostaining. 10-month-old APP/PS1 mice were purchased from SLAC. Quinolinic acid models were constructed by following our previously described protocols [19]. After three weeks of quinolinic acid injection to the striatum, the mouse brains were harvested for staining.

**Human peripheral blood monocyte isolation and culture.** 10 mL of blood were drawn from a healthy male, and the monocytes were isolated by Solarbio Human Blood Monocyte Isolation Kit (P8680, Solarbio, China). The isolated monocytes were plated in 24 well's dishes that contain a poly-o-nithine coated cell culturing slide and then cultured 37°C incubator with 5% $CO_2$.

## Cell line and drug treatment and Flow cytometry

RAW264.7 cells (Applied Biological Materials Inc.) and BV-2 cells (China Center for Type Culture Collection) were maintained in Dulbecco's modified Eagle's medium (DMEM, Gibco, Carlsbad, CA, USA) with 10% fetal bovine serum (FBS, Gibco, Carlsbad, CA, USA) in 37°C/ 5% CO2. Contamination-free cell lines were ensured through testing with a mycoplasma PCR detection kit (BioThrive, Myco-P-20, Shanghai, China). After 5–6 passages, cells were used for experiments.

Human isolated blood monocytes, RAW264.7 and BV-2 cells were separately incubated with IFN-γ, 1-MT, and INCB24360 for 24 hours. The inhibition of IFN-γ-induced IDO-1 was accomplished as follows: After incubating cells with IFN-γ for 12 hours, 1-MT/INCB24360 was added to the culture medium without removing IFN-γ and then incubated for another 12 hours. RAW264.7 cells were separately incubated with MCC950 and oridonin for 24 hours to inhibit NLRP3 inflammasome. The doses of the treatment agents were as follows: IFN-γ, 20 ng/mL (interferon-γ, mouse, Sigma, USA), 1-MT, 20 μM (26988-72-7, Sigma, USA), INCB24360, 20 μM (S7910, Selleck, Shanghai, China), MCC950, 10 μM (inh-MCC, Invivogen, USA), oridonin, 2 μM (Dr. RB Zhou Hefei, China).

After treatments, the cells were digested with 0.25 Trypsin EDTA, stained by PI and a Flow Cytometry (BD FACSCalibur).

## Western blot analysis

The cells were lysed in ice-cold RIPA buffer supplemented with phosphatase inhibitor PMSF for 30 min. The supernatant's protein concentration was measured by a BCA assay kit (PD-BCA-125, BioThrive, Shanghai, China). Proteins (25 μg) were loaded in 12% gels for SDS-PAGE electrophoresis, and then, the proteins were transferred from the gel to PVDF membranes (Millipore, MA, USA). The membranes were blocked with 5% skim milk in TBST (PBS and 0.1% Tween), and incubated at 4°C overnight with antibodies. After incubation with primary antibodies, the membranes were rinsed with TBST 3 times for 5 min each time and then incubated with species-specific horseradish peroxidase-conjugated secondary antibodies (1:5000, Santa Cruz, Germany) for 60 min at room temperature. After 3 washes with TBST for 10 min each time, the membranes were developed with a super-sensitive enhanced chemiluminescence substrate kit (Biothrive Ltd, ECL-P-100, Shanghai, China) for visualization with a Tanon- 4600 imaging system.

The following antibodies were used: rabbit anti-iNOS (1:500, ab3523, Abcam, USA), rabbit anti-CD206 (1:500, BM4881, Boster Biological Technology, CA, USA), rat anti-IDO and rabbit anti-IDO (1:500, 654002, BioLegend, San Diego, CA, USA), rabbit anti-IDO (HPA023149, Thermo Scientific, Lafayette, CO, USA), mouse anti-NLRP3 (1:1000, AG-20B-0014, Adipogen, CA, USA), rabbit anti-caspase-1 (1:500, 3866S, Cell Signaling Technology, MA, USA), rabbit-anti-S6K (polyclonal, AF8962, R&D Systems, MN, USA), rabbit anti-p-S6K (Recombinant Monoclonal Antibody, MAB8963, R&D Systems, MN, USA), anti-α-tubulin (1:10000, HRP-66031, Proteintech, Wuhan, China), and anti-GAPDH (1:10000, HRP-60004, Proteintech, Wuhan, China), Goat anti-IBA-1 antibody (ab5076, Abcam, USA), Rabbit IBA-1 antibody (019–19741, FUJIFILM, VA, USA), CD11b antibody (EPR1344, Cambridge, UK).

## Real-time PCR analysis

EZ-press RNA Purification Kit (B0004DP, USA) was used to extract total RNA by following the manufacturer's protocol. Verso cDNA kit (Thermo Scientific, Lafayette, CO, USA) is applied to cDNA reverse transcription. Quantitative real-time PCR was performed on a Bio-Rad Cx96 Detection System (Bio-Rad, USA) using an SYBR green PCR kit (Applied Biosystems, USA). Primers: 1; iNOS, forward primer (5'-3')GGAGTGACGGCAAACATGACT, reverse primer(5'-3')TCGATGCACAACTGGGTGAAC; 2; CD206, forward primer (5'-3')CTCAACCCAA GGGCTCTTCTAA, reverse primer(5'-3')AGGTGGCCTCTTGAGGTATGTG; 3; TNF-α, forward primer (5'-3')CTGTGAAGGGAATGGGTGTT, reverse primer(5'-3')GGTCACTGTCCCAGCATC TT; 4; Arg-1, forward primer(5'-3')CTCCAAGCCAAAGTCCTTAGAG, reverse primer(5'-3') GGAGCTGTCATTAGGGACATCA; 5; NLRP3, forward primer(5'-3')ATGCTGGCTTCGACATCT CCT, reverse primer(5'-3')GTTTCTGGAGGTTGCAGAGC; 6; Caspase-1, forward primer(5'-3') AGATGCCCACTGCTGATAGG, reverse primer(5'-3')TTGGCACGATTCTCAGCATA; 7; GAPDH, forward primer(5'-3')ATACGGCTACAGCAACAGGG, reverse primer(5'-3') GCCTCTCTTGCTCAGTGTCC.

## Immunofluorescence staining of cell and tissues

Immunostaining was performed as previously described. Briefly, cells were fixed with 4% para-formaldehyde (PFA), incubated with primary antibody overnight, and incubated with fluorescent secondary antibodies and DAPI for 1 hour. After staining, slides were mounted in medium and observed by microscopy. The following antibodies were used: rabbit anti-iNOS (1:200, ab3523, Abcam, USA), rabbit anti-CD206 (1:200, BM4881, Boster Biological Technology, CA, USA), rat anti-F4/80 (1:100, ab16911, Abcam, USA), mouse anti-NLRP3 (1:400, AG-20B-0014, Adipogene, CA, USA), goat and rabbit anti-IBA-1 (1:100, Abcam, USA; 1:1000, FUJIFILM, Japan), rabbit anti-GFAP (1:2000, Abcam, USA). 4′,6-diamidino-2-phenylindole (DAPI, 1:2000, D9542, Sigma, MO, USA). Images were obtained using a Leica SP8 confocal microscope (Leica Microsystems, Japan). Cell counting and morphological analyses were performed using Fiji (ImageJ) software.

## ELISAs

The supernatants of RAW264.7 and BV2 cells were collected to detect the level of interleukin-1β (IL-1β) and interleukin-18 (IL-18). An enzyme-linked immunosorbent assay kit (MeiLian, Shanghai, China) was used according to the manufacturer's instructions. Briefly, the samples were diluted to 1:1 by a standard sample diluent, add 50 μl standard samples and 50 μl detecting samples to the wells, and then supplement the wells with 50 μl of biotin-labeled IL-1β or IL-18 antibodies. After mixing, the wells were incubated at 37°C for 1 h. Discard the liquid, wash the wells for 3 times, add 80 μl streptavidin-conjugated HRP to each well, and incubate at

37˚C for 30 min, rinse the wells with the washing solutions for 3 times. After washing, add 50μl substrate to each well, incubate at 37˚C for 10 min, stop the reaction by 100 μl termination buffer, and read OD values by 450nm.

### Transwell migration assay

Cells were suspended in serum-free DMEM and placed in a 150 μl (E5/ml) cell suspension in the upper chamber of a Transwell apparatus. Then, 800 μl of DMEM containing 10% FBS was added to the lower chamber. After incubating the RAW264.7 cells for 48 h or BV-2 cells for 24 h, the upper Transwell chambers were removed, and the culture medium in the upper chambers was removed by aspiration. The cells in the chambers were fixed with 4% PFA for 15 min, washed three times with PBS, and then stained in the chambers with crystalline violet solution for 30 min. After staining, the upper layer cells were gently swabbed with a cotton swab moistened in PBS, rinsed with PBS three times and dried for use in microscopy.

### Edu essay

Edu assay (Invitrogen) was performed by following the supplier's protocol. After drug treatments, an Edu working solution (10 mM, 1:2000, with green fluorescence) was added to the medium and then incubated for 12 h. After incubation, the cells were fixed with 4% PFA for 20 min, rinsed with PBS, permeabilized with 0.2% Triton X-100 for 20 min, and then added to freshly prepared Click-iT™ reaction solution and incubated for 30 min. After washing, the cells were mounted on slides for visualization by microscopy.

### Imaging and image analysis

A Leica SP8 microscope (Leica, German), Structured illumination microscopy (SIM) (Nikon, Japan) and 880 confocal micros-copy (ZEISS, Jena, Germany) was used to scan all images. ImageJ software (Fiji, NIH, Bethesda, MD, USA) was used to count cells and perform morphological analyses. Imaris 9.5 (Bitplane AG, Zürich, Switzerland) was used for the particle size analysis and figure display.

### Statistical analyses

SPSS version 21.0 (SPSS Inc, Chicago, IL, USA) and GraphPad software were used for data analysis. One-way ANOVA and $t$–test were used to test the differences.

## Results

### Meninges and GBM macrophages/microglia expressed IDO-1 in murine models and humans

To see if the IDO-1+ macrophages/microglia preferentially reside in specific sites in murine model and human brains, we adopted a thick-section staining method to localize IDO-1+ macrophages/microglia in the human brain, human GBM tissues, and the brains from the APP/ PS1 mouse and quinolinic acid models with IBA-1 or GFAP and IDO-1 antibodies. The results showed a few IDO-1+ microglia/macrophages with small bodies and few processes in the brain parenchyma and microglia with larger bodies and longer processes not expressing IDO-1 in the brain parenchyma (Fig 1A-i and 1A-ii). We also observed a considerable number of IDO-1+ microglia/macrophages in the vascular lumen, vascular wall, and perivascular space (Fig 1A-ii) and subarachnoid space in the human brain tissues (Fig 1A-iii and 1B). However, GFAP + astrocytes did not express IDO-1 (Fig 1C). In the tumor region and the region close to tumors, we observed that multiple IDO-1+ microglia/macrophages also closely surrounded

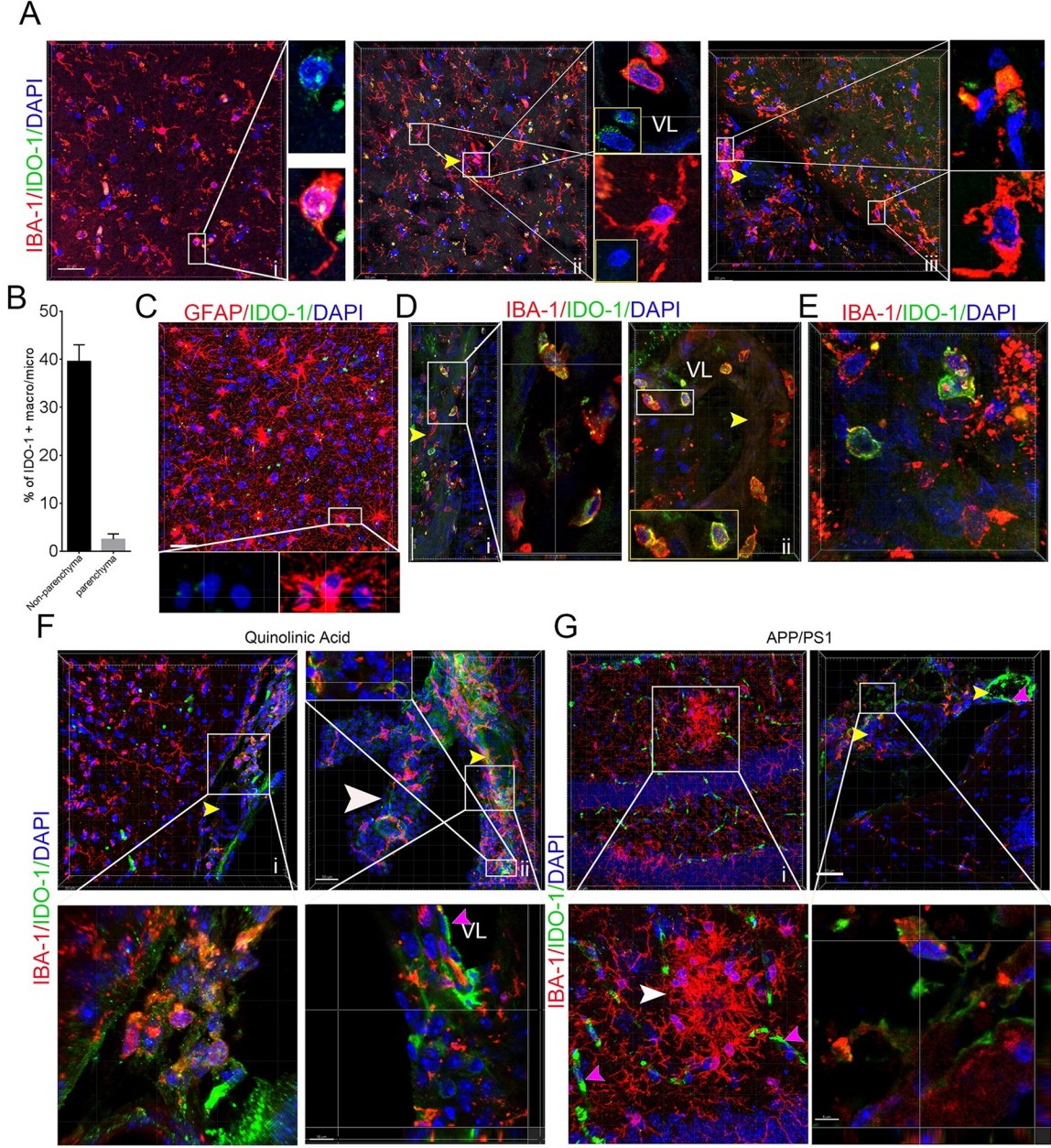

**Fig 1. IDO-1+ macrophages/microglia reside in the meninges and perivascular space of the human and mouse brain and in human GBM tissues. (A)** IBA-1 andIDO-1 antibodies staining in noncancerous brain tissues from GBM patients (i, cortex; ii, cortex with intravascular IDO-1+ cells; iii, the meninges and cortex; yellow arrow, blood vessel). i and ii are from noncancerous parenchyma of a male GBM patient with age 63 and grade IV tumor. iii is from the noncancerous parenchyma of a female GBM patient with age 32 and grade III tumor. **(B)** The percentage of IDO-1+ nonparenchymal macrophages/microglia in the meninges/perivascular space and parenchymal macrophages/microglia in brain (sample numbers, 4). **(C)** IDO-1 and GFAP antibodies staining of noncancerous brain tissues. Noncancerous parenchyma of GBM a female GBM patient with age 55 and grade IV tumor. **(D)** IDO-1 and IBA-1 antibodies staining of the vascular region in the surrounding region of neoplastic lesions. Noncancerous parenchyma of a female GBM patient with age 32 and grade III tumor. **(E)** IDO-1 and IBA-1 staining of GBM tissues. The tumor tissue of a female GBM patient with age 32 and grade III tumor. **(F, G)** IBA-1 and IDO-1 antibodies staining of the microglia of APP/PS1 and quinolinic acid mouse model brains. Boxed area, magnified region. White arrow, choroid plexus. VL, vascular lumen. Pink arrows, endothelial cells.

blood vessels and appeared in the vascular walls or lumen (Fig 1D). Consistent to other observations [20], we also observed that some brain neoplastic cells are highly expressed IDO-1 (Fig 1E). To see IDO-1+ expression patterns in the microglia of murine neurodegenerative models, we stained APP/PS1 mouse and quinolinic acid model brains with anti-IDO-1 and anti-IBA-1 antibodies and found that IDO-1 is not expressed in the microglia of the cortex, hippocampus or striatum in the quinolinic acid injury model brains (Fig 1F-i) or in the activated microglia surrounding amyloid deposits in the APP/PS1 mouse hippocampus and cortex (Fig 1G-i). We observed a significant fraction of microglia in the subarachnoid space and the choroid plexus of the third ventricle highly expressed IDO-1 (Fig 1F-ii and 1G-ii). The vascular endothelial cells, including capillaries and small arteries, expressed IDO-1 in both models. These data implied that IDO-1+ macrophages/microglia might have distinct physiological and immunological properties.

## Inhibiting IDO-1 with INCB24360 reduced most M1 markers but not M2 markers in the RAW264.7 and BV-2 cells

To test how IDO-1 effects macrophage/microglia activities, we treated RAW264.7 and BV-2 cells with 20 ng/mL IFN-γ to increase IDO-1 levels and used 20 μM 1-MT and 20 μM INCB24360 to inhibit IDO-1 activity for 24 h. We detected CD206, an M2 marker, iNOS, an M1 marker, and IDO-1 by western blotting. Consistent with other findings [21], IFN-γ treatment increased IDO-1 and iNOS expression in the RAW264.7 and BV-2 cells (Fig 2A **and** S1A Fig). Inhibiting endogenous IDO-1 with INCB24360 or 1-MT reduced iNOS levels but did not affect CD206 in the RAW264.7 or BV-2 cells, and the effects of INCB24360 were more significant than those of 1-MT (Fig 2A **and** S1A Fig). We also detected M1/M2 markers, including *iNOS*, *TNF-α*, *CD206*, and *Arginase 1* (Arg1) by qRT-PCR, and we found that INCB24360 decreased *iNOS* and *TNF-α* RNA in the RAW264.7 and BV-2 cells (Fig 2B **and** S1B Fig); 1-MT treatment decreased *iNOS* and *TNF-α* RNA levels in the BV2 cells but not in the RAW264.7 cells (Fig 2B **and** S1B Fig). Interestingly, both 1-MT and INCB increased the *CD206* and *Arg-1* RNA levels in the RAW264.7 cells. 1-MT did not affect *CD206* RNA in BV-2 cells; both 1-MT and INCB did not affect the *Arg-1* in BV-2 (Fig 2B **and** S1B Fig). Immunofluorescence staining of RAW264.7 and BV-2 cells treated with INCB24360 and 1-MT revealed that iNOS expression was significantly decreased in INCB24360 groups but CD206 was not changed compared to the levels in the control group (Fig 2C **and** S1C Fig). The data indicate that IDO-1 might induce an increase in M1-like macrophages/microglia in the brain.

## Inhibiting IDO-1 decrease the cellular body and membrane filopodia in RAW264.7 and BV-2 cells

Non-polarized RAW264.7 cells are small and round with few processes [22]. When polarized, RAW264.7 grow large and round with a pancake-like shape, representative of the M1 phenotype, or they become slimmer cells with longer processes, representative of the M2 phenotype [22]. To determine whether iNOS reduction in RAW264.7 and BV-2 cells after INCB24360 treatment caused cellular morphological and filopodium changes, we assessed the morphology of the RAW264.7 and BV-2 cells after IFN-γ, 1-MT, and INCB24360 treatment (Fig 3A and S2A Fig). IFN-γ treatment dramatically increased the number of polarized RAW264.7 and BV-2 cells, while neither the 1-MT nor INCB24360 treatment significantly changed the number of polarized RAW264.7 cells (Fig 3A) or BV-2 cells (S2A Fig). We divided the polarized macrophages into the M1 or M2 types based on their morphology. Consistent with other observations [22], IFN-γ treatment preferentially increased the proportion of M1-like RAW264.7 and BV-2 cells and slightly decreased the proportion of M2 type cells (Fig 3A **and**

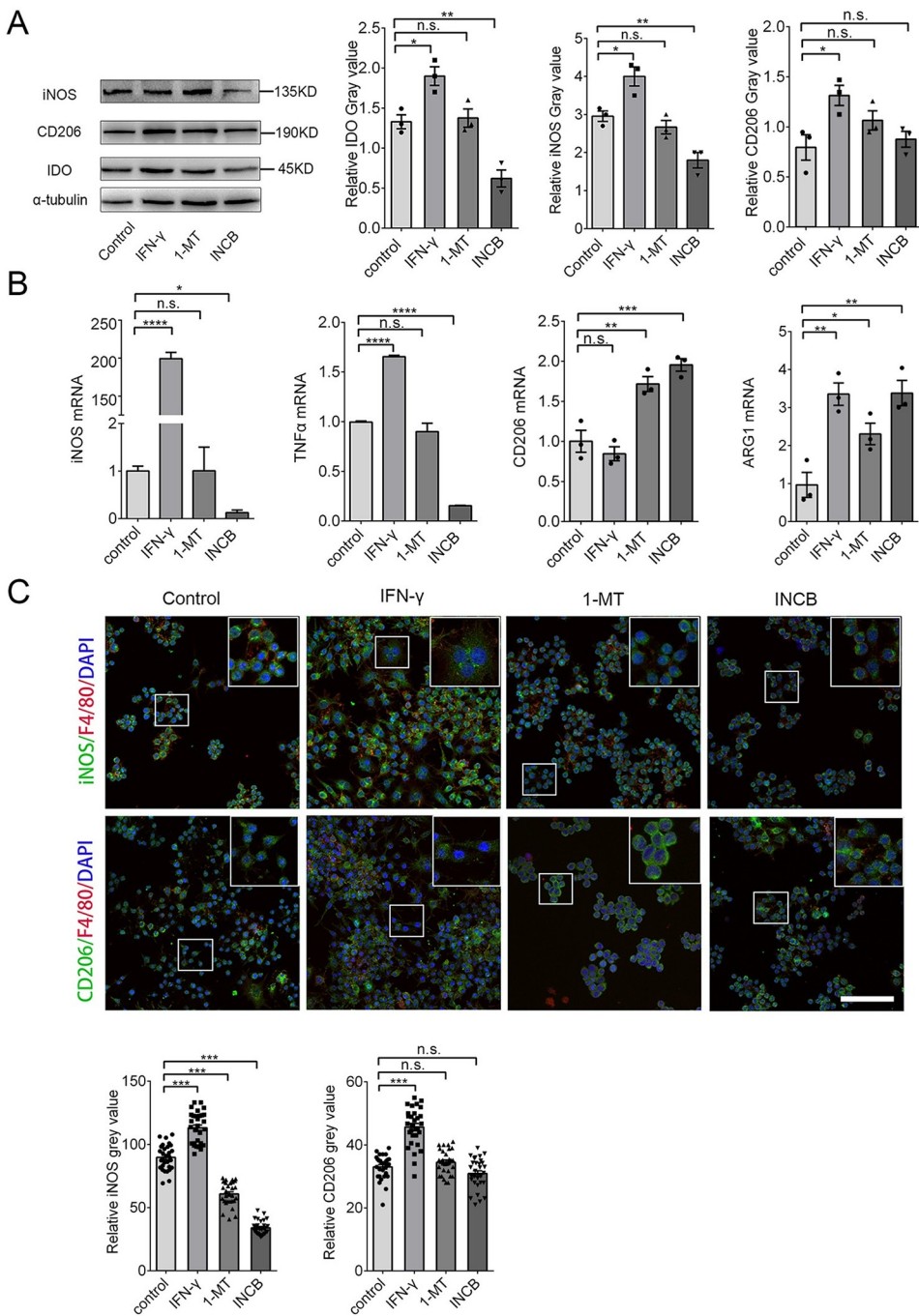

**Fig 2. Inhibition of IDO-1 with 1-MT and INCB24360 decreased iNOS and TNF-α levels. (A)** The IDO-1, iNOS, and CD206 expression in RAW264.7 cells treated with IFN-γ, 1-MT or INCB24360 for 24 h. The relative intensity of IDO-1, iNOS, and CD206 in RAW264.7 cells as measured by ImageJ software. **(B)** The transcription levels of *iNOS*, *TNFα*, *CD206* and *Arg1* in RAW264.7 cells treated with IFN-γ, 1-MT or INCB24360 for 24 h. **(C)** The immunostaining images of iNOS and CD206 in RAW264.7 cells treated by IFN-γ, 1-MT or INCB24360 for 24 h. The relative intensity of iNOS or CD206 in RAW264.7 cells after treatment with IFN-γ, 1-MT or INCB24360, which was measured by ImageJ software. n≥30. Scale bars, 100 μm. One-way ANOVA; all data are expressed as the means ± SEM. *, P<0.05, **, P<0.01; ns, no significant difference.

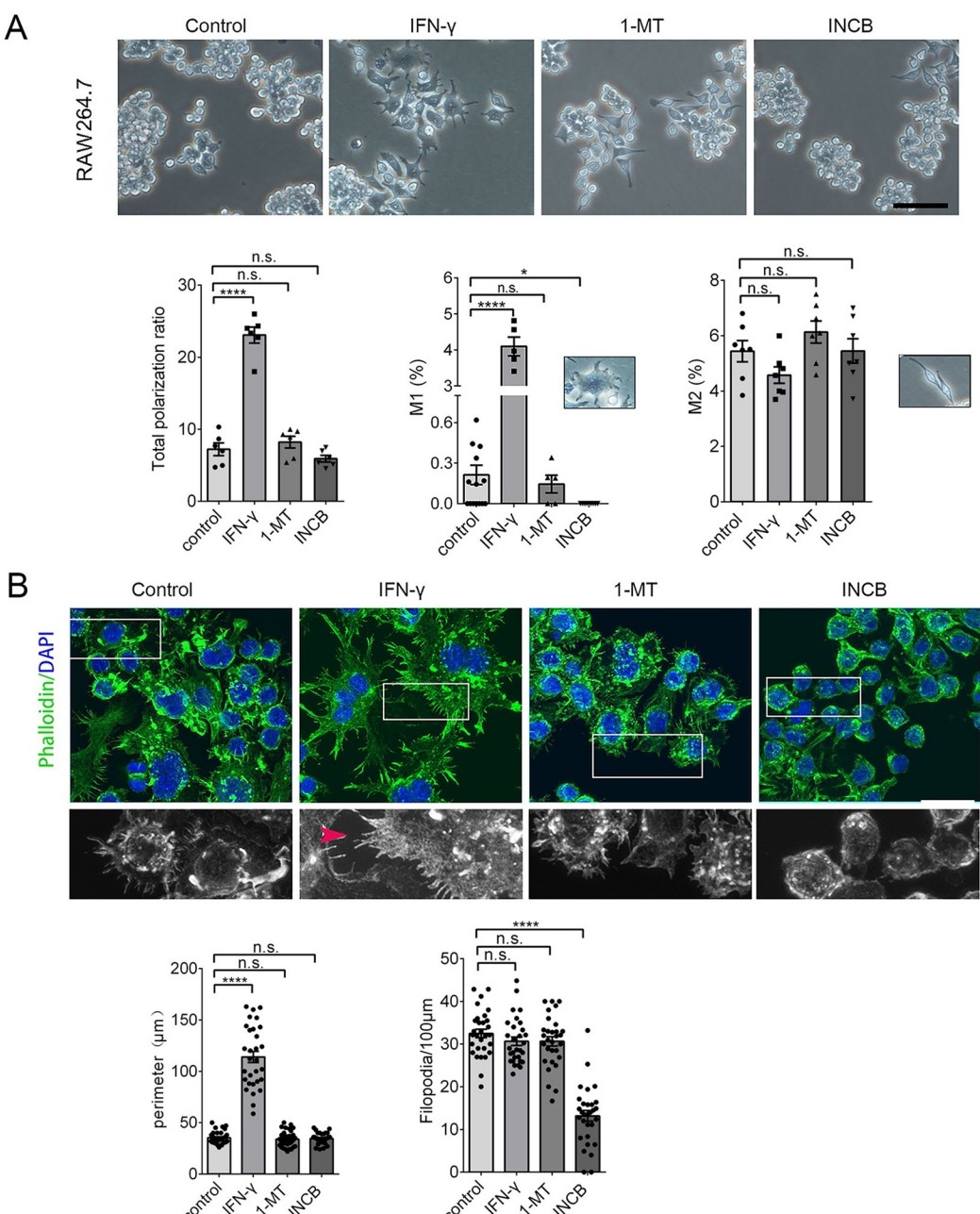

**Fig 3. Inhibiting IDO-1 with 1-MT or INCB24360 reduced the proportion of M1-like but not M2-like RAW264.7 cells.**
(**A**) The typical morphology of RAW264.7 cells treated with IFN-γ, 1-MT and INCB24360 for 24 h. More round and small
RAW264.7 cells were observed in the 1-MT and INCB24360 groups compared to the IFN-γ and control groups. The
percentage of the total polarized, M1-type-cells (ramified), and M2-type-cells (slender) in the control, IFN-γ, 1-MT and
INCB24360 groups. N ≥ 5. Scale bars, 100 μm. (**B**) The phalloidin Alexa-488 staining of RAW264.7 cells treated with IFN-γ,
1-MT or INCB24360 for 24 h. The cellular perimeters in the control, IFN-γ, 1-MT and INCB24360 groups. The density of the
filopodia on the membrane of RAW264.7 cells in the control, IFN-γ, 1-MT and INCB24360 groups. n ≥30. Scale bars, 30 μm.
One-way ANOVA; all data are expressed as the means ± SEM. *, P<0.05, **, P<0.01; ns, no significant difference.

S2A Fig); INCB24360 treatment significantly decreased the proportion of M1 RAW264.7 and
BV-2 cells compared to the control (Fig 3A and S2A Fig). Further, 1-MT treatment did not
change the M1/M2 ratio in the RAW264.7 cells (Fig 3A). However, both 1-MT and
INCB24360 slightly increased the proportion of M2 BV-2 cells (S2A Fig).

M1 macrophages showed multiple filopodia that facilitate the phagocytosis of fluid or foreign substances, including inorganic particles [23]. We treated RAW264.7 and BV-2 cells with IFN-γ, 1-MT, or INCB24360 for 24 h, fixed the cells and stained them with phalloidin Alexa-488, an actin-binding dye, and DAPI. We then observed that the macrophages treated with IFN-γ formed a ruffled border with abundant filopodia (Fig 3B **and** S2B Fig). INCB24360-treated macrophages showed fewer filopodia than the control, IFN-γ, and 1-MT RAW264.7 and BV-2 cell groups (Fig 3B **and** S2B Fig). The cellular perimeter and filopodia density data showed that the 1-MT and INCB24360 treatment did not change the cellular perimeter but decreased the filopodia density on the cellular membrane (Fig 3B **and** S2B Fig). Collectively, these findings implied that inhibition of IDO-1 preferentially blocks cellular size increases and filopodia growth.

## Inhibiting IDO-1 reduces TMR-dextran uptake and phagocytosis of RAW264.7 and BV-2 cells

To determine whether, in addition to a decrease in iNOS and filopodia and the acquisition of a ruffled cellular border, INCB24360 treatment also reduces the phagocytosis ability of the RAW264.7 and BV-2 cells, we administered TMR-dextran to RAW264.7 and BV-2 cells after treating them with IFN-γ, 1-MT, and INCB24360 for 24 h. After 1 h of TMR-dextran treatment, we fixed the cells and counterstained them with phalloidin-Alex-488 and DAPI, and we found that INCB24360 treatment significantly reduced DMR-dextran uptake compared to the amount internalized by the control and IFN-γ+ RAW264.7 cell groups compared to the BV-2 cell groups (Fig 4A–4D).

Macropinocytosis, which depends on membrane ruffling, is the mechanism by which macrophages internalize large amounts of extracellular fluid [24]. The decrease in filopodia on the ruffled border after IDO-1 inhibition implied that IDO-1 might enhance macropinocytosis. The macropinosome ranged from 0.2 μm to 5 μm in diameter [25]. The vesicles, which are larger than 0.75 μm, have been identified as apparent macropinocytic vesicles in DTR-dextran uptake assays performed in some studies [26]. We found that INCB24360 treatment decreased the number of phagocytic vesicles larger than 0.75 μm in RAW264.7 and BV-2 cells (Fig 4E and 4F).

Macrophages, especially tissue-resident macrophages, can phagocytize foreign-derived particulates such as alum and silica. M1 macrophages can also phagocytize foreign-derived particulates [27, 28]. We treated RAW264.7 and BV-2 cells with Latex beads, a spherical polymer particle, and fluorescence red, after incubating the cells with IFN-γ, 1-MT, or INCB24360 for 24 h. After fixing the cells with 4% PFA, we analyzed the phagocytized Latex beads and observed that 1-MT and INCB24360 treatment slightly decrease the Latex beads uptake by the BV-2 cells compared to the control cells, but the decrease was not statistically significant (Fig 4G).

In contrast to M2 macrophages, M1 macrophages show low motility [29, 30]. We used 1-MT and INCB24360 to test their effects on the migration and proliferation of macrophages in a Transwell migration assay and in an Edu staining assay and Flow cytometry to test cell proliferative potential. Consistent with the formation of M1 macrophages, the results showed that IFN-γ treatment dramatically decreased the migration ability of the RAW264.7 and BV-2 cells (S3A Fig) but it not affect the proliferation of the RAW264.7 or BV-2 cells (S3B and S3C Fig); 1-MT and INCB24360 do not affect the migration or proliferation of the RAW264.7 or BV-2 cells (S3A–S3C Fig). Taken together, 1-MT and INCB24360, especially INCB24360, preferentially inhibited the phagocytic ability and macropinocytosis of the macrophages but not their capacities for migration or proliferation.

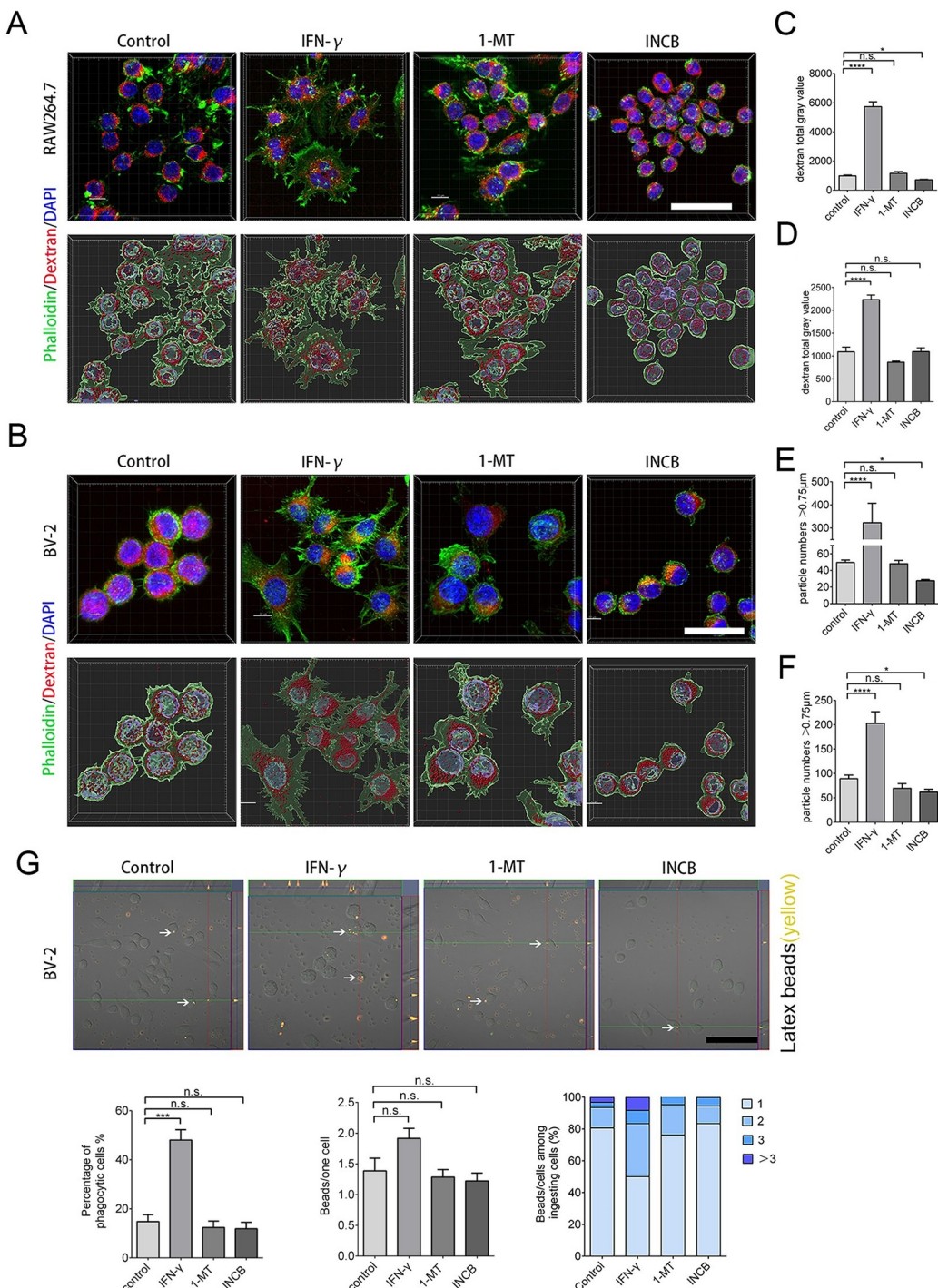

**Fig 4. INCB24360 treatment reduced DTR-dextran uptake by RAW264.7 and BV-2 cells. (A, B)** The typical images of engulfed DTR-dextran in IFN-γ induced RAW264.7 or BV-2 cells after administration of dextran for 1 h in the control, IFN-γ, 1-MT and INCB24360 groups. The phagocytic particles larger than 0.75 μm were analyzed by IMARIS 9.6. Scale bars, 40 μm. **(C, D)** Quantification of DTR-dextran particles in RAW264.7 cells (B) and BV-2 cells (E). Single-cell area X Dextran average grayscale was used as statistical data; n ≥ 15. **(E, F)** DTR-dextran particles larger than 0.75 μm in the control, IFN-γ, 1-MT and INCB24360 groups of IFN-γ induced RAW264.7 cells (C) or BV2 cells (F). n ≥ 15. **(G)** Phagocytosis of latex beads in BV-2 cells treated with IFN-γ, 1-MT or INCB24360 for 24 h. The three statistical charts represent that the phagocytic ratio, the number of beads in each cell, the number of cells with one, two, three, or more beads. Scale bars, 100 μm. One-way ANOVA; all data are expressed as the means ± SEM. *, P<0.05, **, P<0.01; ns, no significant difference.

## Inhibiting IDO-1 with 1-MT or INCB24360 suppresses the IFN-γ-induced iNOS upregulation in RAW264.7 cells

To determine whether IDO-1 inhibitors affected IFN-γ-induced M1 or M2 markers, we treated RAW264.7 and BV-2 cells with IFN-γ for 12 h, and then we added 1-MT or INCB24360 for 24 h. Both 1-MT and INCB24360 significantly decreased iNOS levels but did not change CD206 levels in the IFN-γ-treated RAW264.7 cells; a greater decrease in iNOS was observed in the INCB24360 group (Fig 5A). However, neither 1-MT nor INCB24360 changed the iNOS and CD206 expression in the IFN-γ-treated BV-2 cells (S4A Fig). We also detected *iNOS*, *TNF-α*, *CD206*, and *Arg1* RNA levels by RT-PCR in the four cell groups and found that INCB24360 treatment reduced the *iNOS* and *TNF-α* RNA levels in the IFN-γ-induced RAW264.7 and BV-2 cells (Fig 5B and S4B Fig). INCB24360 treatment significantly increased the transcription level of *CD206* RNA but not *Arg1* RNA in the IFN-γ-treated RAW264.7 cells, but 1-MT treatment did not significantly affect the transcription of *iNOS*, *TNF-α*, *CD206*, or *Arg1* genes (Fig 5B). In agreement with the western blot analysis results, immunofluorescence staining results showed that iNOS expression significantly decreased in the IFN-γ-treated cells treated with INCB24360 or 1-MT compared to the IFN-γ-treated RAW264.7 and BV-2 cells (Fig 5C and S4C Fig). Further, 1-MT treatment significantly increased CD206 expression in the IFN-γ-treated RAW264.7 cells, while 1-MT and INCB24360 increased CD206 expression in the IFN-γ-induced BV-2 cells (Fig 5C **and** S4C Fig).

To determine whether IDO-1 inhibitors also inhibit IFN-γ-induced cellular size and filopodia growth, we treated RAW264.7 and BV-2 cells with IFN-γ for 12 h and then added 1-MT or INCB24360 for 24 h. After imaging, we found that INCB24360 treatment not only decreased the total number of polarized RAW264.7 cells (Fig 6A) and BV-2 cells driven by IFN-γ (S5A Fig), but INCB24360 treatment preferentially decreased the number of IFN-γ-induced M1 RAW264.7 and BV-2 cells (Fig 6A **and** S5A Fig). Interestingly, INCB24360 treatment significantly increased the IFN-γ-induced M2 macrophage proportion (Fig 6A **and** S5A Fig). Further, 1-MT treatment decreased the number of total IFN-γ-induced polarized macrophages, and this decrease affected M1 or M2 macrophages equally (Fig 6A **and** S5A Fig). The findings support the supposition that INCB24360 can reverse IFN-γ-driven M1 macrophage increases.

To determine whether IDO-1 inhibitors block IFN-γ-driven ruffled border formation and filopodia growth, we treated RAW264.7 and BV-2 cells with IFN-γ for 12 h, and then added 1-MT or INCB24360 for 24 h, fixed the cells and stained them with phalloidin Alexa-488 and DAPI. Our data showed that INCB24360 inhibited the formation of the IFN-γ-induced ruffled border and filopodia formation in the RAW264.7 and BV-2 cells (Fig 6B and S5B Fig). Measuring and counting data showed that both 1-MT and INCB24360 treatments significantly inhibited the increase in RAW264.7 cell size induced by IFN-γ (Fig 6B) but did not affect BV-2 cells (S5B Fig) INCB24360 treatment also inhibited the increase in filopodia density on the cellular membrane induced by IFN-γ (Fig 6B **and** S5B Fig), but 1-MT treatment did not restrict the increase in filopodia density on the cellular membrane induced by IFN-γ (Fig 6B **and** S5B Fig). Inhibiting IDO-1 with INCB24360 or 1-MT reduced both cell size and filopodia growth in the macrophages/microglia.

## Inhibiting IDO-1 suppresses the IFN-γ induced endocytic, macropinocytic and phagocytic abilities of RAW264.7 and BV-2 cells

To determine whether 1-MT and INCB24360 can decrease the endocytic and macropinocytic ability induced by IFN-γ, we treated freshly isolated human peripheral blood monocytes, BV-2 and RAW264.7 cells with IFN-γ for 12 h and then treated them with 1-MT and

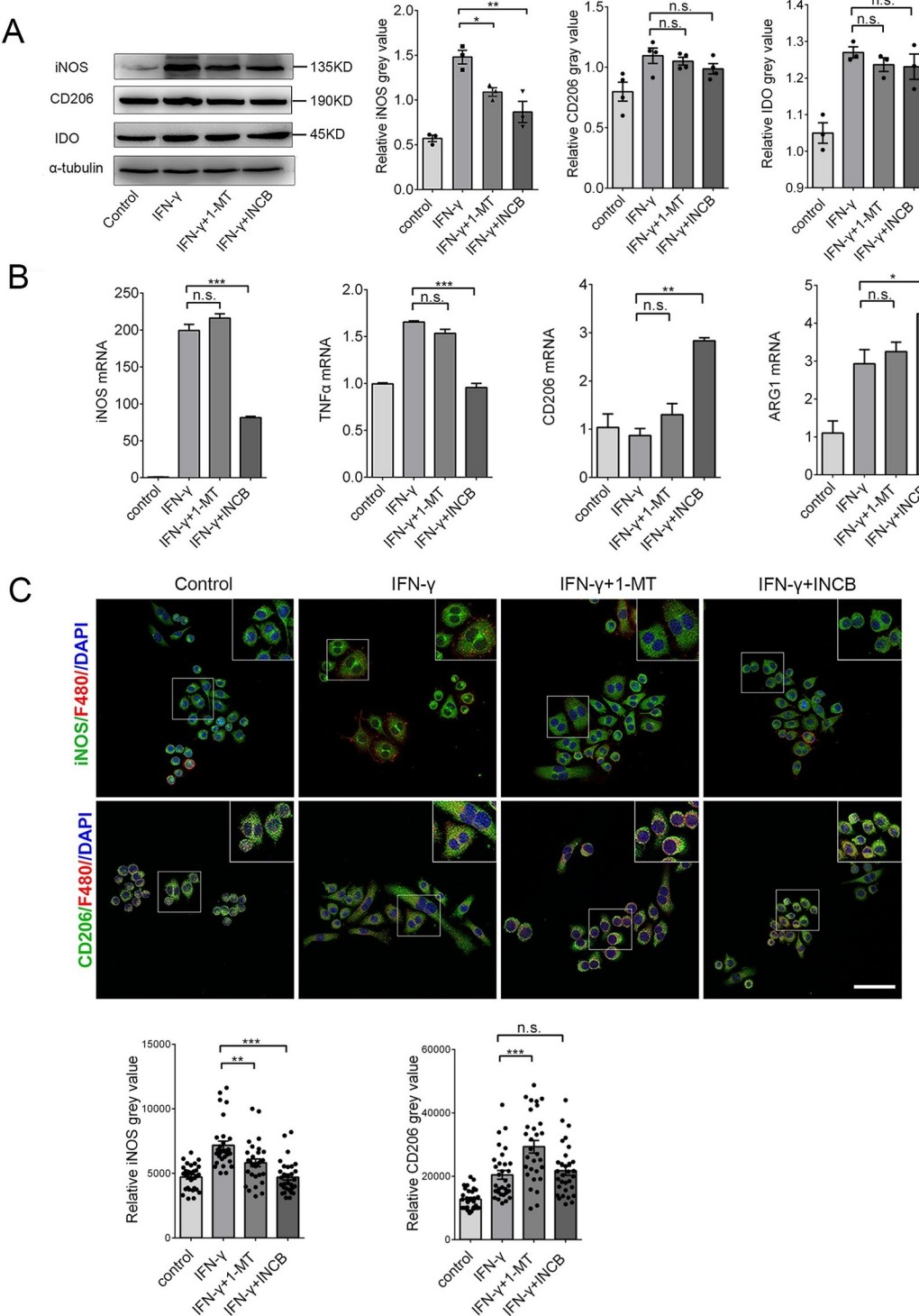

**Fig 5. Inhibiting IDO-1 with INCB24360 or 1-MT suppresses IFN-γ induced iNOS and TNFα increases in RAW264.7 cells.** (**A**) iNOS, CD206 and IDO-1 expression in RAW264.7 cells treated with IFN-γ, IFN-γ+1-MT or IFN-γ + INCB24360 for 24 h (12 h with IFN-γ, following another 12 hours with 1-MT or INCB24360) and the relative intensity of iNOS, CD206, and IDO-1 in the RAW264.7 cells as measured by ImageJ. (**B**) The transcription levels of *iNOS*, *TNFα*, *CD206* and *Arg1* in RAW264.7 cells treated with IFN-γ, IFN-γ+1-MT or IFN-γ + INCB24360 for 24 h. (**C**) The immunostained images of iNOS and CD206 in RAW264.7 cells treated by IFN-γ, IFN-γ+1-MT or IFN-γ + INCB24360 for 24 h. The quantification of iNOS and CD206 intensity in RAW264.7 cells as measured by ImageJ software. Scale bars, 50 μm. One-way ANOVA; all data are expressed as the means ± SEM. *, P<0.05, **, P<0.01; ns, no significant difference.

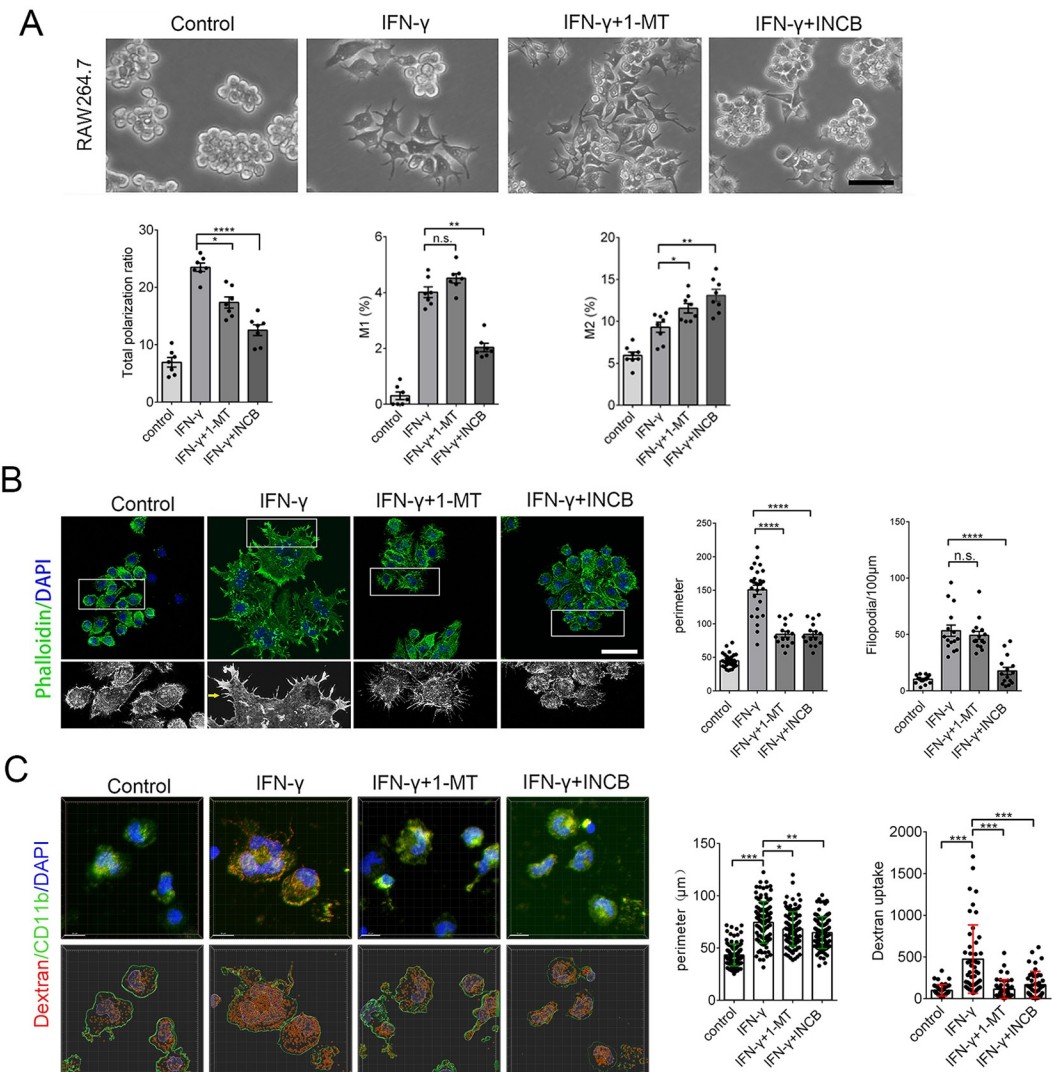

**Fig 6. Inhibiting IDO-1 with INCB24360 reduced the M1-like macrophage proportion in the RAW264.7 cells induced by IFN-γ. (A)** The typical morphology of RAW264.7 cells after treatment with IFN-γ, IFN-γ+1-MT and IFN-γ + INCB24360. A greater number of round and small RAW264.7 cells were observed in the IFN-γ+1-MT and IFN-γ+INCB24360 groups compared to the IFN-γ group. The percentage of the polarized M1-like macrophages (ramified) and M2-like macrophages (slender) in the control, IFN-γ, IFN-γ+1-MT and IFN-γ + INCB24360 groups. N ≥ 5. Scale bars, 60 μm. **(B)** The phalloidin Alexa-488 staining of RAW264.7 cells treated with IFN-γ, IFN-γ+1-MT and IFN-γ + INCB24360 for 24 h. The cellular perimeters in the control, IFN-γ, IFN-γ+1-MT and IFN-γ + INCB24360 groups. The density of the filopodia of RAW264.7 cell in the control, IFN-γ, IFN-γ+1-MT and IFN-γ + INCB24360 groups. One-way ANOVA; all data are expressed as the means ± SEM. *, P<0.05, **, P<0.01; ns, no significant difference. Scale bars, 40 μm. **(C)** INCB24360 and 1-MT decreased IFN-γ induced cellular size increase and dextran uptake. Human peripheral blood monocytes of 47 healthy male was isolated by Solarbio Human Peripheral Monocyte Isolation Kit, and cells were treated with IFN-γ, IFN-γ+1-MT and IFN-γ + INCB24360 for 24 h. High-resolution images were taken by Zeiss 880 Airscan Microscopy after staining with CD11B antibody. Endocytic dextran particles were analyzed by Spot of Imaris 9.7. Images in each group, n>10; Cellular perimeter analysis, cells>100; dextran particles analysis, cells>40.

INCB24360 for another 12 h. We then added DTR-dextran to the culture medium. After treating with DTR-dextran for 1 h, we fixed the cells and counterstained them with phalloidin Alex-48h. In human blood monocytes, both INCB24360 and I-MT decreased IFN-γ-induced cellular size increase and the TMR-dextran uptake ability (Fig 6C). INCB24360

decreased the TMR-dextran uptake ability of both the RAW264.7 and BV-2 cells; 1-MT significantly decreased the TMR-dextran uptake ability of the RAW264.7 cells but not of the BV-2 cells (Fig 7A–7D). We analyzed the DTR-dextran phagocytic vesicles that were larger than 0.75 μm, which is reflective of macropinocytosis, by Imaris9.6 software. The findings indicated that INCB24360 treatment decreased the number of vesicles larger than 0.75 μm phagocytosed by RAW264.7 and BV-2 cells, and 1-MT significantly decreased the number of vesicles larger than 0.75 μm phagocytosed by RAW264.7 cells but not by BV-2 cells (Fig 7E and 7F).

To determine the effect of 1-MT and INCB24360 on macrophage phagocytic ability, we treated RAW264.7 and BV-2 with IFN-γ for 12 h. We added 1-MT or INCB24360 and incubated the cells for 24 h, then introduced Latex beads to the culture medium. After fixing the cells, we analyzed the number of phagocytized Latex beads in the cells and found that INCB24360 and 1-MT treatment significantly decreased the number of Latex beads phagocytosed by BV-2 cells. The number of latex beads phagocytosed by IFN-γ-induced BV-2 cells treated with INCB24360 was significantly lower than that phagocytosed by the untreated IFN-γ-induced BV-2 cells. However, the number of latex beads phagocytosed by IFN-γ-induced BV-2 cell treated with 1-MT was not significantly lower than that phagocytosed by untreated IFN-γ-induced BV-2 cells (Fig 7G). In summary, I-MT and INCB24360, but especially INCB24360, inhibited the endocytic, phagocytic, and macropinocytic ability of IFN-γ-treated macrophages.

## Inhibiting IDO-1 with 1-MT and INCB24360 reduces NLRP3 expression in RAW264.7 and BV-2 cells

The NLRP3 inflammasome, a large protein complex, includes NLRP3, ASC, and caspase 1 [31]. The NLRP3 inflammasome is involved in M1 macrophage formation [32]. Curcumin, an IDO-1 inhibitor, suppresses NLRP3 expression in chronic unpredictable mild stress (CUMS) and reduces depression-like behaviors [33]. To determine whether NLRP3 and IDO-1 are co-upregulated in macrophages/microglia in vivo, we stained the human brain tissues with anti-IBA-1, anti-IDO-1 and anti-NLRP3 antibodies and observed that IDO-1+ macrophage/microglia in the perivascular space (Fig 8A). To determine how IDO-1 affects NLRP3 expression, we treated RAW264.7 and BV-2 cells with IFN-γ, 1-MT, and INCB24360 for 24 h. We detected NLRP3 expression by western blotting and found that both 1-MT and INCB24360 decreased NLRP3 but not caspase-1 expression in the RAW264.7 and BV2 cells (Fig 8B). The decreases in NLRP3 were statistically significant in the INCB24360 and 1-MT groups (Fig 8B). We also used immunostaining of anti-NLRP3 and anti-iNOS antibodies to detect changes in NLRP3 and iNOS expression. Consistent with immunoblotting and RT-PCR results, INCB24360 treatment significantly decreased NLRP3 expression in the RAW264.7 cells (Fig 8C) and BV-2 cells (S6C Fig).

To determine whether 1-MT and INCB24360 can also decrease IFN-γ-induced NLRP3 expression, we also treated RAW264.7 cells with IFN-γ for 12 h, and then added 1-MT or INCB24360 for 24 h to inhibit IDO-1 activity. The data showed that both 1-MT and INCB24360 inhibited IFN-γ-induced NLRP3 expression but not caspase-1 expression (Fig 8D). The RT-PCR data also showed that INCB24360 and 1-MT reduced IFN-γ-induced *NLRP3* gene transcription in both RAW264.7 and BV-2 cells but did not reduce *caspase-1* transcription (Fig 8E **and** S6F Fig). We immunostained these treated cells with anti-NLRP3 and anti-iNOS antibodies, and the results of the immunofluorescence staining of RAW264.7 cells were consistent with the protein blotting results (Fig 8F **and** S6F Fig).

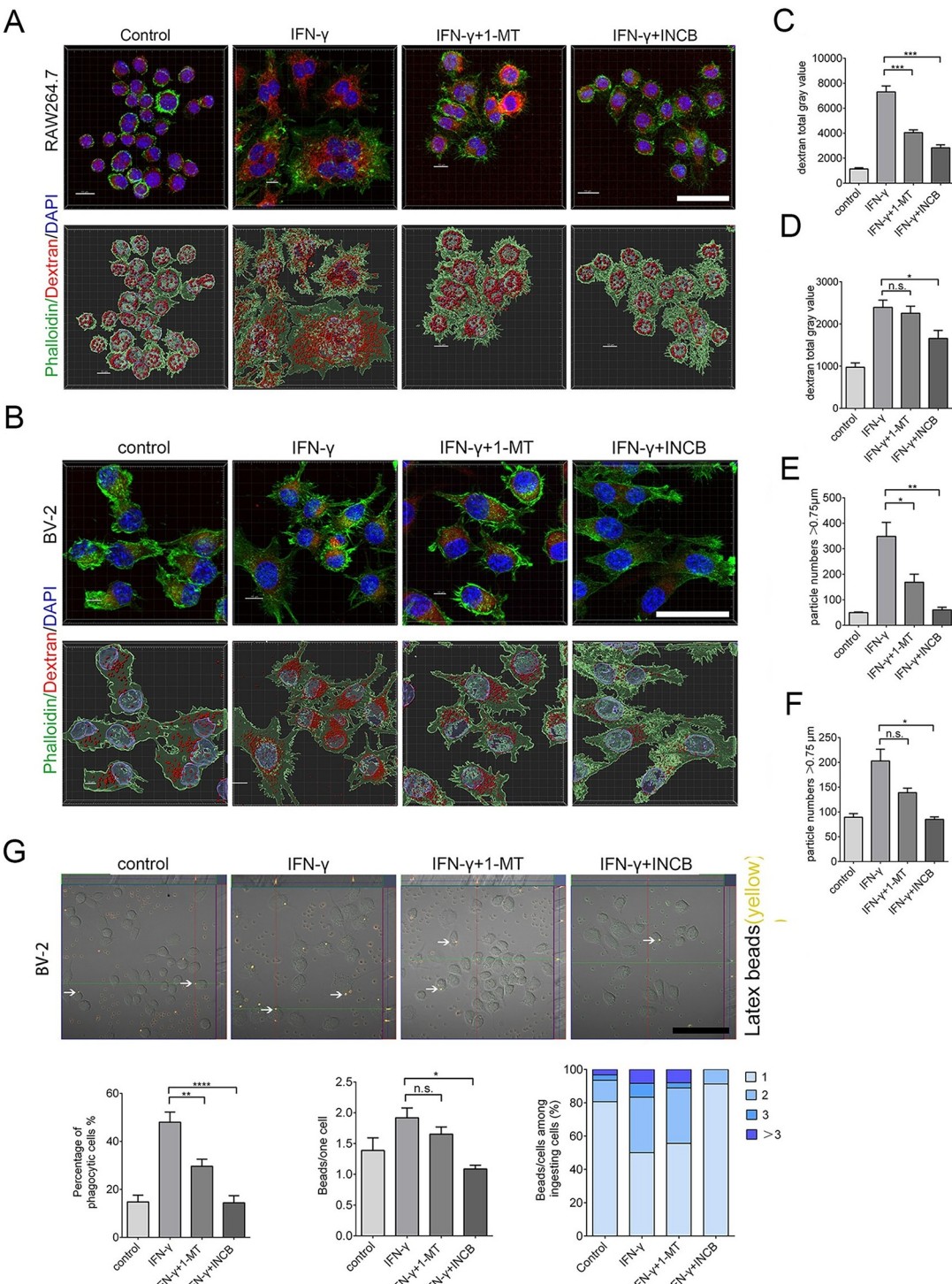

**Fig 7. Inhibiting IDO-1 with 1-MT or INCB24360 reduced DTR-dextran uptake by RAW264.7 and BV-2 cells induced by IFN-γ. (A, B)** The typical images of engulfed DTR-dextran in IFN-γ induced RAW264.7 (A) or BV-2 (B) cells after administration of dextran for 1 h in the control, IFN-γ, IFN-γ+1-MT and IFN-γ + INCB24360 groups. The phagocytic particles larger than 0.75 μm were analyzed by IMARIS 9.6. Scale bars, 40 μm. **(C, D)** Quantification of DTR-dextran particles in RAW264.7 cells (C) and BV-2 cells (D). Single-cell area X Dextran average grayscale was used as statistical data; n ≥ 15. **(E, F)** DTR-dextran particles larger than 0.75 μm in the control, IFN-γ, IFN-γ+1-MT and IFN-γ + INCB24360 groups of IFN-γ induced RAW264.7 cells (E) or BV2 cells (F). n ≥ 15. **(G)** Phagocytosis of latex beads in BV-2 cells treated with IFN-γ, IFN-γ +1-MT or IFN-γ + INCB24360 for 24 h. The three statistical charts represent that the phagocytic ratio, the number of beads in each cell, the number of cells with one, two, three, or more beads. Scale bars, 100 μm. One-way ANOVA; all data are expressed as the means ± SEM. *, P<0.05, **, P<0.01; ns, no significant difference.

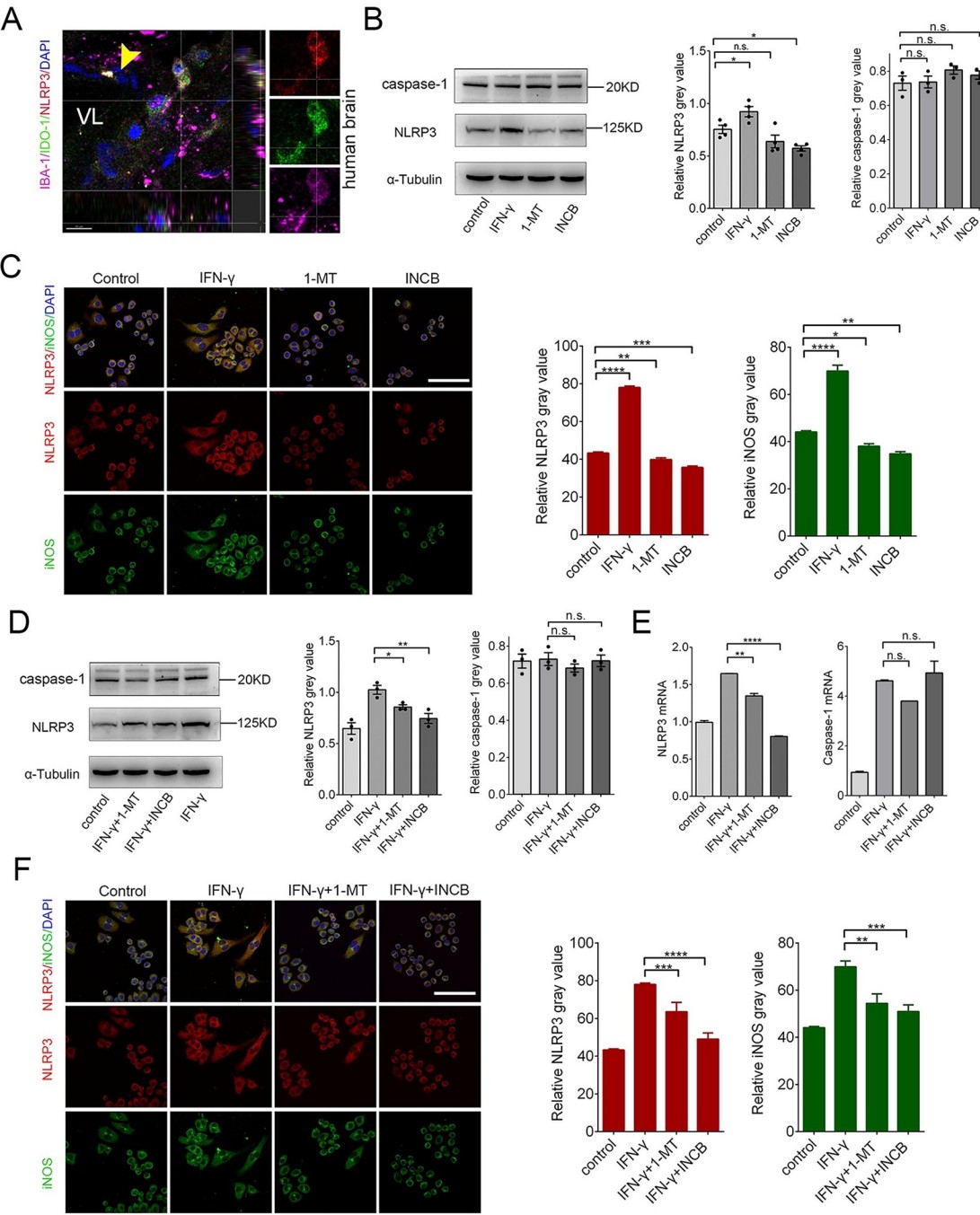

**Fig 8. Inhibiting IDO-1 with 1-MT or INCB24360 reduced NLRP3 expression in RAW264.7 cells. (A)** NLRP3, IDO-1 and IBA-1 triple-staining in the meninges of a human noncancerous brain (Yellow arrow, vessels; VL, vascular lumen). **(B)** The changes of NLRP3 and caspase-1 in RAW264.7 cells treated with IFN-γ, 1-MT or INCB24360 for 24 h. The relative intensity of NLRP3 and caspase-1 in RAW264.7 cells as measured by ImageJ software. **(C)** The immunostained results of NLRP3 and iNOS in RAW264.7 cells treated with IFN-γ, 1-MT or INCB24360 for 24 h. The relative NLRP3 or iNOS intensity in RAW264.7 cells as measured by ImageJ software. n≥10. Scale bars, 80 μm. **(D)** The changes of NLRP3 and caspase-1 in RAW264.7 cells treated with IFN-γ, IFN-γ +1-MT or IFN-γ + INCB24360 for 24 h. The relative intensity of NLRP3, caspase-1 in RAW264.7 cells as measured by ImageJ software. **(E)** The transcription levels of *NLRP3* and *caspase-1* in RAW264.7 cells after treatment with IFN-γ, IFN-γ+1-MT or IFN-γ + INCB24360 for 24 h. **(F)** The immunostained images of NLRP3 and iNOS in RAW264.7 cells treated with IFN-γ, IFN-γ+1-MT or IFN-γ + INCB24360 for 24 h. The relative NLRP3 or iNOS intensity in RAW264.7 cell as measured by ImageJ software. Scale bars, 80 μm. One-way ANOVA; all data are expressed as the means ± SEM. *, P<0.05, **, P<0.01; ns, no significant difference. Scale bars, 80 μm.

## 1-MT and INCB24360 treatments increase IL-1β secretion in RAW264.7 and BV-2 cells

The NLRP3 inflammasome activates caspase 1, which cleaves pro-IL-1β and pro- IL-18, and leads to the secretion of IL-1β and IL-18 [31]. Inhibiting IDO with 1-MT dramatically caused a surge in IFN-γ, IL-1β, IL-6, and TNF-α secretions from the macrophages infected with the influenza virus [9, 16]. To determine whether the decrease in NLRP3 by 1-MT and INCB24360 affects IL-1β and IL-18 secretion, we treated RAW264.7 cells with 20 ng/mL IFN-γ, 20 μM 1-MT, and 20 μM INCB24360 for 24 h. We detected IL-1β and IL-18 secretion levels in the medium by ELISA and found that IL-1β and IL-18 levels in the culture medium of IFN-γ-, 1-MT-, and INCB24360-treated RAW264.7 and BV-2 cells were significantly increased (S7A and S7B Fig).

Tryptophan deprivation by IFN-γ-induced IDO-1 inhibits mTORC1 kinase, and tryptophan or 1-MT treatment reversed mTORC1 inhibition [2, 34]. To determine whether INCB24360 reverses mTORC1 inhibition, we used an anti- phosphorylated (p) S6K (T389) antibody to evaluate mTORC1 activation levels in RAW264.7 cells. Interestingly, we did not observe a direct effect of 1-MT or INCB24360 treatment on pS6K (T389) levels in the RAW264.7 cells (S7C Fig). We observed that inhibiting IFN-γ-induced IDO-1 with 1-MT and INCB24360 increased IL-1β and IL-18 secretion by RAW264.7 and BV-2 cells (S7A and S7B Fig). The increase in IL-1β secretion in the INCB24360-treated RAW264.7 and BV-2 cells induced by IFN-γ was significantly higher than that in the RAW264.7 and BV-2 cells induced with IFN-γ alone (S7B Fig). Interestingly, 1-MT significantly increased IL-18 secretions in IFN-γ-induced RAW264.7 cells by but not IL-1β (S7A and S7B Fig). Additionally, in agreement with other findings [27, 34], inhibiting IFN-γ-induced RAW264.7 cells with 1-MT or INCB24360 dramatically increased pS6K (T389) levels (S7D Fig). Activation of mTORC1 by 1-MT and INCB24360 in IFN-γ-induced cells might partially explain why 1-MT and INCB24360 increase IL-1β secretion by RAW264.7 cells.

NLRP3 deficiency inhibits the IDO-1 upregulation induced by LPS in the hippocampal microglia [35]. To determine whether NLRP3 inhibition affects endogenous IDO-1 levels in RAW264.7 cells, we suppressed NLRP3 in RAW264.7 cells with MCC950, an inhibitor of NLRP3 by targeting ATP-hydrolysis [36], and oridonin, a covalent NLRP3 inhibitor [37]. After exposing RAW264.7 cells to MCC950 or oridonin for 24 h, we detected NLRP3 and IDO-1 expression (S7E Fig), and found that inhibiting NLRP3 with oridonin and MCC950 increased IDO-1 expression in the RAW264.7 cells (S7F Fig). Our data indicate that NLRP3 and IDO-1 might co-regulate each other in macrophages.

## Discussion

IDO-1 expression in placental epithelial cells, neoplastic cells, and macrophage reject T-cell infiltrates and diminishes the immune response [38, 39]. We showed that IDO-1+ macrophages/microglia reside in the perivascular and subarachnoid spaces at the brain parenchyma interface. Using IDO-1 inhibitors, we found that IDO-1 enhanced macrophage/microglia endocytic, phagocytic, and macropinocytic capacities via increases in cell size and filopodia growth. Inhibition of IDO-1 in macrophages/microglia reduced NLRP3 expression but increased the secretion of IL-1β. Consistent with a lower migrating ability of M1 macrophages and morphometric analysis [29], IFN-γ treatment significantly reduced the migration of RAW264.7 and BV-2 but not 1-MT, INCB24360. The multiple filopodia of macrophage in IFN-γ treatment might contribute to phagocytic or endocytic ability but not migrating. Our findings showed that IDO-1+ macrophages/microglia have strong endocytic, phagocytic, and macropinocytic capacities and weaker proinflammatory properties in the meninges.

Macrophages/microglia scavenge cellular debris, effete cells, invading microbes, and metabolites in the cerebrospinal fluid (CSF) and parenchyma to maintain homeostasis [5, 7]. The meninges, perivascular, and choroid-plexus macrophages patrol the CSF to surveil waste and invasion [5, 7]. The observation of vascular wall-traversing or vascular luminal IDO-1+ macrophages indicates that some IDO-1+ macrophages might originate from a monocyte, which is consistent with the transcriptome database on macrophages/microglia [18]. In contrast to the meninges with abundant immune infiltrates, T cells are almost absent in healthy brain parenchyma. T-cell infiltration in brain parenchyma is a hallmark of multiple sclerosis (MS) [40]. Ectopic T-cell infiltration is also observed in the parenchyma of Parkinson's and Alzheimer's patients [41–43]. The only pathway to parenchyma for T cells involves crossing the blood-brain barrier (BBB) and the pia matter. Thus, whether IDO-1+ macrophages/microglia partially or significantly participate in the rejection of T cells at the brain parenchyma is worth exploring in the future.

Consistent with attenuated ruffled border and filopodia formation, inhibiting IDO-1 decreased the endocytic, macropinocytic, and phagocytic abilities of macrophages/microglia. Surprisingly, we also observed that IDO-1+ macrophages in metastatic lymph nodes and tumors possess multiple processes and larger body sizes in pancreatic cancer patients **(unpublished data)**. These observations further support the supposition that IDO-1 is related to large cell size and greater macrophage/microglia phagocytic and macropinocytic abilities. The decrease in endocytic, macropinocytic, and phagocytic ability and iNOS and TNF-α levels after inhibiting IDO-1 showed that IDO-1 drives the formation of M1-like macrophages. IL-1β is a critical proinflammatory cytokine in M1 macrophages [44]. Previous data showed that IDO-1$^{-/-}$ mice suppressed LP-BM5 replication by secreting excessive type I IFN, and 1-MT treatment dramatically led to a surge in IL-1β secretion by macrophages in conjunction with virus-induced inflammation [9, 16, 17]. The enhancement of IL-1β secretion after inhibiting IDO-1 hints at an anti-inflammatory role for IDO-1 in macrophages/microglia. IDO-1 is a downstream enzyme of the NLRP3 inflammasome, and IDO-1 upregulation induced by lipopolysaccharide (LPS) is diminished in the glial cells of *Nlrp3$^{-/-}$* mice, including microglia [32, 35]. The inhibition of IDO-1 with curcumin decreased NLRP3 expression [33]. We showed that both 1-MT and INCB24360 reduced endogenous or IFN-γ-induced NLRP3 expression, and inhibiting NLRP3 also increased IDO-1 expression, indicating that the relationship of NLRP3 and IDO-1 is bidirectional, and the direction depends on the activation status of macrophages/microglia.

In the future, to what extent IDO-1 expressing microglia/macrophages contribute to immune barriers of the brain parenchyma and if the IDO-1 expressing endothelial cells also contribute to the T cell rejection by parenchyma are worth exploring. Another interesting point is that if IDO-1 deregulation in macrophage/microglia correlated with brain autoimmunity diseases caused by the abnormal entrance of T cells to parenchyma, such as Multiple Sclerosis [45].

Our findings collectively show that IDO-1+ macrophages/microglia in meninges are phagocytes with higher scavenging ability and lower proinflammatory activity, implying that IDO-1+ macrophages/microglia combined with IDO-1 + endothelial cells might be involved in the prevent T cells from the meninges/perivascular space from entering the parenchyma or crossing the BBB and may prevent the overactivation of immune response.

## Supporting information

**S1 Fig. Inhibition of IDO-1 in BV-2 with 1-MT and INCB 24360 decreased iNOS and TNF-α levels.** (A) The iNOS and CD206 expression in BV-2 cells treated with IFN-γ, 1-MT or

INCB24360 for 24 h. (B) The transcription levels of iNOS, TNFα, CD206 and Arg1 in BV-2 cells treated with IFN-γ, 1-MT or INCB24360 for 24 h. (C) The immunostaining images of iNOS and CD206 in BV-2 cells treated with IFN-γ, 1-MT or INCB24360 for 24 h. The relative intensity of iNOS or CD206 in BV-2 cells after treatment with IFN-γ, 1-MT or INCB24360, which was measured by ImageJ software. n≥20. Scale bars, 100μm. One-way ANOVA; all data are expressed as the mean ± SEM. *, P<0.05, **, P<0.01; ns, no statistical difference. (TIF)

**S2 Fig. 1-MT and INCB-24360 treatment reduced M1-like macrophage while increased M2-like macrophage in BV-2. (A)** The typical morphology of BV-2 cells treated with IFN-γ, 1-MT and INCB24360 for 24 h. The percentage of M1-like macrophage (ramified); M2-like macrophage (slender) in the control, IFN-γ, 1-MT and INCB24360 groups. N ≥ 5. Scale bars, 80μm. **(B)** The phalloidin Alexa-488 staining of BV-2 cells treated with IFN-γ, 1-MT or INCB24360 for 24 h. The cellular perimeters in the control, IFN-γ, 1-MT and INCB24360 groups. The density of the filopodia on the membrane of BV-2 cells in the control, IFN-γ, 1-MT and INCB24360 groups. n ≥10. Scale bars, 40μm. One-way ANOVA; all data are expressed as the mean ± SEM. *, P<0.05, **, P<0.01; ns, no statistical difference. (TIF)

**S3 Fig. 1-MT, and INCB treatment did not change the migrating and proliferating capacity of RAW264.7 and BV-2. (A)** The representative images of RAW264.7 or BV-2 cells treated with IFN-γ, 1-MT or INCB24360 in Transwell assay. Scale bars, 150μm. Quantifying migrating RAW264.7 or BV2 cells in Transwell assay. N = 3. Counts were done in ImageJ software. **(B)** The representative images of EDU assays in RAW264.7 or BV-2 cells treated with IFN-γ, 1-MT or INCB24360 for 24 h. Scale bars, 150μm. The percentage of the Edu- positive RAW264.7 or BV-2 cells treated with IFN-γ, 1-MT or INCB24360 for 24 h. N = 3, repeats. Count were done by ImageJ software. Scale bars, 150μm. One-way ANOVA; all data are expressed as the mean ± SEM. *, P<0.05, **, P<0.01; ns, no statistical difference. **(C)** The cell cycles of RAW264.7 cells treated with IFN-γ, 1-MT or INCB24360 for 24 h by Flow cytometry after PI staining. N = 4, repeats. (TIF)

**S4 Fig. Inhibiting IDO-1 with INCB24360 suppresses IFN-γ induced iNOS and TNFα increases in BV-2 cells. (A)** iNOS and CD206 expression in BV-2 cells treated with IFN-γ, IFN-γ +1-MT or IFN-γ +INCB24360 for 24 h. **(B)** The transcription levels of iNOS and TNFα in BV-2 cells after treating with IFN-γ, IFN-γ +1-MT or IFN-γ +INCB24360 for 24 h. **(C)** The immunostaining images of iNOS and CD206 in BV-2 cells treated with IFN-γ, IFN-γ +1-MT or IFN-γ +INCB24360 for 24 h. iNOS or CD206 intensity measured by ImageJ software. Scale bars, 100μm. One-way ANOVA; all data are expressed as the mean ± SEM. *, P<0.05, **, P<0.01; ns, no statistical difference. (TIF)

**S5 Fig. 1-MT and INCB-24360 treatment reduced M1-like macrophage while increased M2-like macrophage in IFN-γ induced BV-2. (A)** The typical morphology of BV-2 cells after treatment with IFN-γ, IFN-γ+1-MT and IFN-γ + INCB24360; the percentage M1-like macrophage (ramified), and M2- like macrophage (slender) in the IFN-γ, IFN-γ +1-MT and IFN-γ +INCB24360 groups. N ≥ 5. Scale bars, 100μm. **(B)** The phalloidin Alexa-488 staining of BV2 cells treated with IFN-γ, IFN-γ+1-MT and IFN-γ + INCB24360 for 24 h. The cellular perimeters in the control, IFN-γ, IFN-γ+1-MT and IFN-γ +INCB24360 groups. The density of filopodia of BV2 cells in the control, IFN-γ, IFN-γ +1-MT and IFN-γ + INCB24360 groups. n≥10. Scale bars, 50μm. One-way ANOVA; all data are expressed as the mean ± SEM. *, P<0.05, **,

P<0.01; ns, no statistical difference.
(TIF)

**S6 Fig. 1-MT and INCB24360 treatment reduced NLRP3 expression and NLRP3 gene transcription in BV-2. (A)** NLRP3 and caspase-1 expression in BV-2 cells after treated with IFN-γ, 1-MT and INCB for 24 h. **(B)** The transcription levels of NLRP3 and caspase-1 in BV-2 cells treated with IFN-γ, 1-MT and INCB for 24 h. **(C)** The immunostaining images of NLRP3 and iNOS in BV-2 cells treated with IFN-γ, 1-MT and INCB24360 for 24 h. NLRP3 or iNOS intensity measured by ImageJ. n≥20. Scale bars, 50μm. **(D)** NLRP3 and caspase-1 expression in BV-2 cells treated with IFN-γ, IFN-γ +1-MT or IFN-γ +INCB24360 for 24 h. **(E)** The transcription levels of NLRP3 and caspase-1 in BV-2 cells treated with IFN-γ, IFN-γ +1-MT or IFN-γ +INCB24360 for 24 h. **(F)** The immunostaining images of NLRP3 and iNOS in BV-2 cells treated with IFN-γ, IFN-γ +1-MT or IFN-γ +INCB for 24 h. NLRP3 or iNOS intensity measured by ImageJ. Scale bars, 50μm. One-way ANOVA; all data are expressed as the mean ± SEM. *, P<0.05, **, P<0.01; ns, no statistical difference.
(TIF)

**S7 Fig. 1-MT and INCB24360 enhance IL-1β secretion in BV-2. (A)** IL-1β and IL18 secretion Levels in RAW264.7 cells (with ELISA) treated with IFN-γ, 1-MT or INCB24360 for 24 h. IL-1β and IL18 secretion Levels in RAW264.7 cells (with ELISA) treated with IFN-γ, IFN-γ +1-MT or IFN-γ +INCB24360 for 24 h. **(B)** IL-1β and IL18 secretion Levels in BV2 cells (with ELISA) treated with IFN-γ, 1- MT or INCB24360 for 24 h. IL-1β and IL18 secretion Levels in RAW264.7 cells (with ELISA) treated with IFN-γ, IFN-γ+1-MT or IFN-γ +INCB24360 for 24 h. **(C, D)** S6K and p-S6K protein levels in RAW264.7 treated with IFN-γ, 1-MT or INCB24360 for 24 h. S6K and p-S6K protein levels in RAW264.7 treated with IFN-γ, IFN-γ+1-MT or IFN-γ +INCB24360 for 24 h. **(E)** The changes of NLRP3 and IDO expression in RAW264.7 treated by MCC950 and Oridonin for 24 h. **(F)** The representative immunostaining results of NLRP3 and IDO RAW264.7 cells treated with MCC950 and IDO for 24 h. The relative levels of NLRP3 or IDO intensity in RAW264.7 cells after drug treatment, measured by image J. Scale bars, 50μm. One-way ANOVA; all data are expressed as the mean ± SEM. *, P<0.05, **, P<0.01; ns, no statistical difference.
(TIF)

**S1 Raw images.**
(PDF)

## Acknowledgments

We thank everyone who supported this project and opened their facilities to us after the Covid-19 outbreak so that we could finish this work.

## Author Contributions

**Conceptualization:** Rong Ji, Hexige Saiyin.

**Data curation:** Rong Ji, Xinyu Chen, Renqiang Sun.

**Formal analysis:** Rong Ji.

**Funding acquisition:** Wenshi Wei.

**Investigation:** Rong Ji, Lixiang Ma, Li Zhang, Hexige Saiyin, Wenshi Wei.

**Methodology:** Rong Ji, Lixiang Ma, Hexige Saiyin.

**Project administration:** Wenshi Wei.

**Resources:** Lixiang Ma.

**Software:** Rong Ji, Xinyu Chen, Hexige Saiyin.

**Supervision:** Lixiang Ma, Wenshi Wei.

**Writing – original draft:** Rong Ji.

**Writing – review & editing:** Lixiang Ma, Hexige Saiyin.

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
