## [Decision Letter · Decision Letter 0]

26 Apr 2021

PONE-D-21-06075

Characterization of IDO-1 expressing macrophages/microglia in the meninges and perivascular space of the human and murine brain

PLOS ONE

Dear Dr. Saiyin

Thank you for submitting your manuscript to PLOS ONE. After careful consideration, we feel that it has merit but does not fully meet PLOS ONE’s publication criteria as it currently stands. Therefore, we invite you to submit a revised version of the manuscript that addresses the points raised during the review process.

We look forward to receiving your revised manuscript.

Kind regards,

Nagaraj Kerur

Academic Editor

PLOS ONE

Journal Requirements:

Additional Editor Comments:

Dear Dr. Saiyin,

Thank you for your patience while the manuscript was being evaluated by the reviewers. We now have evaluation by three reviews. As you see below the reviewer have unanimously recognized the significance of the work presented, however several serious technical and conceptual concerns have been raised. We ask you to address the reviewers comments, particularly those related to quality of the immunofluorescence data. All immunofluorescence images should be supported by appropriate positive and negative controls by including isotype antibody staining. Additional the manuscript needs significant improvement in narrative as pointed out by the reviewers. It would be helpful to present your work and describe your data in the context of existing literature about IDO-1 in brain microglia.

Additionally, RAW264.7 and BV-2 cells are not god surrogate for brain microglia cells, studies in these cells should reproduced in primary mouse/human microglia.

Reviewers' comments:

Reviewer's Responses to Questions

**Comments to the Author**

1. Is the manuscript technically sound, and do the data support the conclusions?

Reviewer #1: Yes

Reviewer #2: Partly

Reviewer #3: Yes

2. Has the statistical analysis been performed appropriately and rigorously? 

Reviewer #1: Yes

Reviewer #2: No

Reviewer #3: Yes

3. Have the authors made all data underlying the findings in their manuscript fully available?

Reviewer #1: Yes

Reviewer #2: Yes

Reviewer #3: Yes

4. Is the manuscript presented in an intelligible fashion and written in standard English?

Reviewer #1: Yes

Reviewer #2: Yes

Reviewer #3: Yes

5. Review Comments to the Author

Reviewer #1: The manuscript by Rong Ji and colleagues titled “Characterization of IDO-1 expressing macrophages/microglia in the meninges and perivascular space of the human and murine brain” described the role of Indoleamine 2,3-dioxygenase 1 (IDO-1) enzyme in supporting the activities of the macrophages/microglia in meninges using human and murine brain samples. The authors were also explained the physiological and immunological role of IDO-1+ macrophages/microglia after treating with 1-MT and INCB2436, inhibitory molecules in murine and human models using immunofluorescence staining, cell migration and proliferation assays. The real time PCR and western blot analysis were used to support the data. If the author would have added other cell proliferation assays and flow cytometry data (if available), that would have been more effective to prove their hypothesis. The manuscript may be appropriate for the publication after a major revision and answering the following comments.

Major comments:

A)Abstract-

1. In the line 3- says” Indoleamine 2,3-dioxygenase 1 (IDO-1) expressed in macrophages rejects T-cells”. It is not clear

how the IDO-1 rejects T-cells? It would be good if the author explains with some more examples.

2. Line 6: disease models- please specify which diseases models are you mentioning here?

3. Line 10: 1-MT and INCB2436- It would be convenient for the readers if the author states clearly about these inhibitory

molecules.

B) Introduction-

4. it is specified that “limitans are thought to contribute to T-cell rejection”. It is not clear is it immune mediated T-cell

rejection or inhibition of migration of T-cells. How do you explain the T-cells rejection? Hypothesis may be reformed

clearly.

5. The sentence “IDO-1 does not drive the formation of M1 macrophages” contradicts your own statements which says IDO

is expressed in M1s. It would be more clear if the author could rewrite the introduction with suitable references.

C) Materials and Methods:

6. Samples details from the human GBM patients are not available. In order to analyze in all possible aspects, more

information on the age or gender of the human patients may be included.

7. The passage numbers and the source/ATCC equivalent details of the cell lines RAW264.7 and BV-2 cells may be

included.

8. Real time PCR: please specify what is the limit of detection, or limit of quantification and the PCR efficacy of the Real

time RT-PCR used in the study.

9. The procedure for Interleukin ELISA is missing. Brief procedures for the ELISA may be included.

10. Please specify the statistical tests used in the study in the method section.

D) Results and discussion:

11. In the Fig legends the human and mouse brain samples were not clearly specified. Please specify the tissue of origin.

12. In Fig legend 1. Explanation for Fig 1E is missing. Is it the magnified region of Fig 1D? is it neoplastic lesions or normal

brain tissues?

13. Have you tested the migration and proliferation of both M1 and M2 macrophages along with 1-MT and INCB2436

treatment? If so discuss the results in comparison with each types.

14. The possible explanation on the differential effects of 1-MT, INCB24360, and IFN-g treatments on proliferation and

migration ability of the M1 or M2 macrophages need an elaborated in discussion.

15. The discussions need to be re written adding more references on the functional effect of the IDO-1. At the same time

the shortcomings of the study and the future prospects of your study need to be mentioned.

Minor comments:

1. In abstract - Line 15: IDO-1 expression in perivascular macrophages instead of IDO-1 in macrophages.

2. There are some lines in the text where IDO-1+ has been mentioned, it would be convenient to mention IDO-1

expressing cells.

3. In Fig 2 the units of the graphs need to be mentioned as fold change or log change.

4. Abstract Line 7: “Here” - Please specify is this at the steady state? Human or mouse?

5. The primer details of the house keeping gene, alpha-tubulin is missing.

Reviewer #2: The current study presents intriguing finding in the brain immunology. The authors propose that IDO-1+ macrophages/microglia in meninges have higher scavenging ability which might be involved in preventing meninges/perivascular space T cells from entering the parenchyma maintaining immunosuppressive environment in the brain. While the manuscript represents an interesting finding and has a decent amount of immune fluorescence evidences and data, there are major considerations that prevents the manuscript’s publication in its current form.

Major Considerations:

1- The major issue in this manuscript is the lack of enough quantitative evidences to support the data. The main analysis in the study was based on manual morphometric analysis of the immune fluorescent photos. While this is a valid analysis, it is subjective and require further verification with other studies “when cells are manually selected, a heavy operator‐dependent bias is introduced, impairing procedure reproducibility (Ruffinatti, Genova et al. 2020)”.

• Fig.1A, the control photo shows extensive microglia/macrophage (almost 1:2 microglia/neuron) raising a question about the health of the brain tested and quality of the preparation (high background noise).

• Fig 1C, the authors need to show higher magnification of the cells without the red filter similar to the others.

• Fig. 1E, (Assume GBM brain, no legend for) IDO-1+ cells are negative for IBA-1. How the authors explain these findings? Although, it is reported that macrophages form about 30-50% of inflammatory infiltrate in the tumor mass and they are known to be IDO+?(Sevenich 2018)

• Fig.3A, the authors measured the control cells’ filopodia. While they choose the upper cells to measure, another type of cells with extensive processes could be visualized in the same filed. What is the rationale that authors used in selecting the cells for analysis?

• The authors need to add the negative control for the staining to exclude non specific staining, for example the neurons in the same field should be negative for both staining.

• Fig. 3A, the authors concluded on the polarization ratio between M1&M2 using morphology as a differentiating point. While there is reported difference in the morphology between M1&M2, the inverted phase morphology is not enough as the accuracy of the morphological differentiation between the two cells is based on differences in size, perimeter, shape, intensity, and texture of the actin and nuclear stain (Rostam, Reynolds et al. 2017), which wouldn’t be achieved by the inverted phase light microscope. Accordingly, other methods like phenotypic marker expression (Calprotectin, MR), cytokines analysis should be used for differentiation the two types. Also, quantitative measurement of the M1/M2 ratio using pan-macrophage markers staining and analysis by flowcytometry would have given more precise data (Antonios, Yao et al. 2013).

• Fig. 7, the authors are reporting on the dextran uptake and latex beads phagocytosis. While the photos are representative, we suggest integration of the results with flowcytometry uptake studies.

• Fig.2&5, while the cytokines’ mRNA expression is a valid representative way, including supernatant’s protein analysis using ELISA, similar to supplementary figure 7, could add to the results.

2- The statistics needs to be revised thoroughly through the study, for example:

• Fig.2B (mRNA expression level of CD206, Arg.1), there is a visible difference in the mean between control and 1-MT, yet it does not show statistical significance. This could be explained by small sample size, or unequal variability between the groups or non-normal distribution of the data. Accordingly, optimum sample size >6, blotting of the individual values and confirming the validity of the statistical test can solve this.

• Fig.3A, How the authors explain the variability between the sample size between groups in the same comparison? And how One way ANOVA was conducted with missing values? (control N= 7, while the IFN n=5, IMT n=4), while the legend states that n=7. This again was observed in Fig.3B, Fig.5A and Fig.6A.

• How many times the experiments were repeated?

• At figure 6A, the authors stated that n>5 without stating what is the sample size exactly and if there is discrepancy between groups. While this is a valid way to represent sample size, however if you are going to statistically compare between them using One Way ANOVA, the exact sample size in each group should be reported. This again was observed in Fig. 4,5 and 7.

To solve the previous issues, meticulous revision needs to be done to the figures and the legends. I also recommend using dot plot figure whenever possible instead of bar graph to show the individual values and adding the mean and the SEM values in the text specially with the polarization ratios.

3- A previous study in 2012 has found that “IDO Expression in Brain Tumors Increases the Recruitment of Regulatory T Cells and Negatively Impacts Survival”. Based on this study findings, how the authors can explain this? (Wainwright, Balyasnikova et al. 2012)

Minor Considerations:

1- Please review the references (1-4) as they do not match the text in the first paragraph. While the text mentions on IDOs, these references describe the microglia role in Parkinson’s disease and the polarization of macrophage.

2- In the legend of figure 1, the authors need to describe which is murine brain and which is human brain as the figures should be able to stand by itself.

3- Figure 1E is not explained either in the results section nor the figure legend.

4- Fig. 2, the statistical significance was not demonstrated in some graphs.

5- The manuscript needs to be revised for typo error like

• Page 17 line 14, However However (delete one of them)

• Page14 line 20, Adipogenic? (Should be Adipogen?)

References:

Antonios, J. K., Z. Yao, C. Li, A. J. Rao and S. B. Goodman (2013). "Macrophage polarization in response to wear particles in vitro." Cellular & Molecular Immunology 10(6): 471-482.

Rostam, H. M., P. M. Reynolds, M. R. Alexander, N. Gadegaard and A. M. Ghaemmaghami (2017). "Image based Machine Learning for identification of macrophage subsets." Scientific Reports 7(1): 3521.

Ruffinatti, F. A., T. Genova, F. Mussano and L. Munaron (2020). "MORPHEUS: An automated tool for unbiased and reproducible cell morphometry." Journal of Cellular Physiology 235(12): 10110-10115.

Sevenich, L. (2018). "Brain-Resident Microglia and Blood-Borne Macrophages Orchestrate Central Nervous System Inflammation in Neurodegenerative Disorders and Brain Cancer." Frontiers in Immunology 9(697).

Wainwright, D. A., I. V. Balyasnikova, A. L. Chang, A. U. Ahmed, K.-S. Moon, B. Auffinger, A. L. Tobias, Y. Han and M. S. Lesniak (2012). "IDO Expression in Brain Tumors Increases the Recruitment of Regulatory T Cells and Negatively Impacts Survival." Clinical Cancer Research 18(22): 6110.

Reviewer #3: In this manuscript Ji et al investigated differential expression of IDO1 in microglia cells in the brain paranchyma, meninges, brain tumors and brain injury models. In addition, they also investigated potential function of IDO1 in microglia/macrophages. Below are some of the concerns that need to be addressed.

1. Throughout the manuscript, please improve the writing. Logically explain why an experiment was done and what is the conclusion

2. Rephrase the concluding paragraph in the introduction- it is confusing and hard to understand.

3. Please explain why INF-Gamma was used- it is kind of choppy and it appears out of nowhere.

4. Please include genetic inhibition of IDO1 and carry out some of the phenotypic data to provide additional evidence for IDO1 functions in microglia.

6. PLOS authors have the option to publish the peer review history of their article (what does this mean?). If published, this will include your full peer review and any attached files.

Reviewer #1: No

Reviewer #2: No

Reviewer #3: No

---

## [Author Response · Author response to Decision Letter 0]

7 Jun 2021

Response to editor-Dr. Nagaraj Kerur and reviewers

Dear Dr. Saiyin,

Thank you for your patience while the manuscript was being evaluated by the reviewers. We now have evaluation by three reviews. As you see below the reviewer have unanimously recognized the significance of the work presented, however several serious technical and conceptual concerns have been raised. We ask you to address the reviewers’ comments, particularly those related to quality of the immunofluorescence data. All immunofluorescence images should be supported by appropriate positive and negative controls by including isotype antibody staining. Additional the manuscript needs significant improvement in narrative as pointed out by the reviewers. It would be helpful to present your work and describe your data in the context of existing literature about IDO-1 in brain microglia. Additionally, RAW264.7 and BV-2 cells are not good surrogate for brain microglia cells, studies in these cells should reproduced in primary mouse/human microglia.

Response to the editor: Thanks for your kind comments. All staining has positive and negative control. As we are from a medical university, we are good at immunostaining and quality controls of staining(Han et al., 2021; Hexige et al., 2015; Ma et al., 2012). We have selected qualified and well-tested antibodies during staining. In addition, we have used multiple human tissues to testify the specificity of our antibodies. IDO-1 antibody is tested in pancreatic cancer tissues and liver cancer, and multiple cell lines. The IDO-1 staining patterns in pancreatic cancer are consistent with other works in pancreatic cancer and immunohistochemistry database (Blair et al., 2019) (https://www.mybiosource.com/monoclonal-human-antibody/ido-1/303376). iNOS, CD206 that we used in this paper have plenty of publications to support the specificity of these antibodies both in immunostaining and western blotting analysis. Our immunostaining system has a clean background in mouse brain (Reviewer and Editor only Fig.4A).

You are right. RAW264.7 and BV-2 cells are not good surrogates for brain microglia. BV-2 is negative for IBA-1 staining and only partially represents microglia/macrophages' biological and immunological behaviors. However, the relative undifferentiated status of these two cell lines [round shape with fewer processes] makes them valuable to test the biological effects of drugs or chemokines. Using human peripheral blood monocytes, we found that the behaviors of RAW264.7 under IFN-γ treatment resemble human blood-derived monocytes [The cellular size and phagocytic abilities increased]. The difference is that RAW264.7 and BV-2 formed more filopodia than human monocyte in IFN-γ treatment. Thus we have selected the two cells to test the effects of IDO-1 and its inhibitors. In the revised version, we have used blood-derived monocytes to test the IDO-1 effects on macrophages' phagocytic abilities and morphological phenotype [Refer Fig.6C]. 

In addition, we observed that IDO-1 positive macrophages in the metastatic lesion or invasive pancreatic cancers are characterized with larger body size and more processes compared to IDO-1 low or negative cells (please refer to editor and reviewer only figure 2). These observations further support our findings in this work.

Response to the reviewer

Comments to the Author

Reviewer #1: The manuscript by Rong Ji and colleagues titled "Characterization of IDO-1 expressing macrophages/microglia in the meninges and perivascular space of the human and murine brain" described the role of Indoleamine 2,3-dioxygenase 1 (IDO-1) enzyme in supporting the activities of the macrophages/microglia in meninges using human and murine brain samples. The authors have also explained the physiological and immunological role of IDO-1+ macrophages/microglia after treating with 1-MT and INCB2436, inhibitory molecules in murine and human models using immunofluorescence staining, cell migration, and proliferation assays. The real time PCR and western blot analysis were used to support the data. If the author would have added other cell proliferation assays and flow cytometry data (if available), that would have been more effective to prove their hypothesis. The manuscript may be appropriate for the publication after a major revision and answering the following comments.

Thanks for raising these questions. We found that IFN-γ treatment dramatically increased the cellular size. 15-25% of IFN-γ treated cells reached 50-80µm in diameter, some of them reached 100µm in diameters. We have consulted our core facility technician, and they informed us that the size of NOZZLE of Flowcytometry in the core facility is 100µm. The general rule of thumb is that your nozzle size should be about 4-5 times larger than the size of the cells being interrogated. Theoretically, our flow cytometry only analyzes the cell that is smaller than 33µm in diameters. If we analyze a cell with 50-80µm in diameters, we need a specialized NOZZLE. Unfortunately, our core facility is not equipped with a NOZZLE larger than 100µm. Thus we have used staining to evaluate the cells.

Major comments:

A) Abstract-

1. In the line 3- says" Indoleamine 2,3-dioxygenase 1 (IDO-1) expressed in macrophages rejects T-cells". It is not clear how the IDO-1 rejects T-cells? It would be good if the author explains with some more examples.

Response 1-1: Thanks for pointing this out. To limit the space, we have shortened the sentence. IDO-1 metabolizes tryptophan into kynurenine, resulting in depleting tryptophan and producing immune-suppressive kynurenine that recruits Tregs and MDSC. Tregs and MDSC suppressed cytotoxic T cell activity. We have shortened the sentence in the abstract. 

2. Line 6: disease models- please specify which diseases models are you mentioning here?

Response 1-2: Thanks for pointing this out. We specified disease models in the revised version.

3. Line 10: 1-MT and INCB2436- It would be convenient for the readers if the author clearly states these inhibitory molecules.

Response 1-3: Thanks for pointing this out. We have clearly stated that in the revised version.

B) Introduction-4. it is specified that “limitans are thought to contribute to T-cell rejection”. It is not clear is it immune mediated T-cell rejection or inhibition of migration of T-cells. How do you explain the T-cells rejection? Hypothesis may be reformed clearly.

Response 1-4: Thanks for pointing this out. The question that you have raised is in the scale of our future work. Based on current understanding, glia limitation is a physical barrier that bars the T cells into the parenchyma. However, this explanation did not include the possibility of the existence of immune cell roles. In this work, we discuss the physiological and immunological roles of IDO-1+ macrophage/microglia in vitro and to what extent IDO-1 expression affects the behaviors of macrophages. In the future, we wish to explore the roles of IDO-1+ macrophage/microglia. 

5. The sentence “IDO-1 does not drive the formation of M1 macrophages” contradicts your own statements which says IDO is expressed in M1s. It would be more clear if the author could rewrite the introduction with suitable references.

Response 1-5: Thanks for pointing this out. This is an inadvertent mistake. We have deleted it

C) Materials and Methods:

6. Samples details from the human GBM patients are not available. In order to analyze in all possible aspects, more information on the age or gender of the human patients may be included.

Response 1-6: We have done IF in 4 patients. We have added the patients’ information in supplementary materials (Please refer to supplementary table-1). As the size is smaller than 10, we thought that the analysis of relationships between clinicopathological characteristics and IDO-1 is difficult to provide insightful information. 

7. The passage numbers and the source/ATCC equivalent details of the cell lines RAW264.7 and BV-2 cells may be included.

Response 1-7: Thanks for pointing this out. After 5-6 passages, the RAW264.7 and BV-2 cells were used for experiments. The RAW264.7 cells were purchased from Applied Biological Materials Inc. (T9096, Richmond BC, Canada). The BV-2 cells were purchased from China Center for Type Culture Collection. 

8. Real time PCR: please specify what is the limit of detection, or limit of quantification and the PCR efficacy of the Real time RT-PCR used in the study.

Response 1-8: Thanks for raising this question. We thought that the limit of detection is not the scope of our study. We all know that transcription levels of one gene correlated with protein levels sometimes in cells or tissues; other times, the transcription levels of one gene do not reflect the protein levels in cells or tissue. Thus we used IF and WB to strengthen our arguments. 

9. The procedure for Interleukin ELISA is missing. Brief procedures for the ELISA may be included. 

Response 1-9: Thanks. We have added this part to the materials and method section in the revised version (please refer to the method and materials). 

10. Please specify the statistical tests used in the study in the method section.

Response 1-10: Thanks for pointing this out. We have specified the statistical tests in the revised version.

D) Results and discussion:

11. In the Fig legends the human and mouse brain samples were not clearly specified. Please specify the tissue of origin. 

Response 1-11: Thanks. We have specified the origin of tissue in the revised version.

12. In Fig legend 1. Explanation for Fig 1E is missing. Is it the magnified region of Fig 1D? is it neoplastic lesions or normal brain tissues? 

Response 1-12. This is an inadvertent labeling mistake. Sorry for bring confusion. We corrected it.

13. Have you tested the migration and proliferation of both M1 and M2 macrophages along with 1-MT and INCB2436 treatment? If so discuss the results in comparison with each types.

Response 1-13: Thanks for pointing this out. We did not test the migration and proliferation of both M1 and M2 macrophages along with 1-MT and INCB2436 treatment. The migration and proliferation of both M1 and M2 macrophages in 1-MT and INCB2436 treatment not the prominent scope of this work.

14. The possible explanation on the differential effects of 1-MT, INCB24360, and IFN-g treatments on proliferation and migration ability of the M1 or M2 macrophages need an elaborated in discussion. 

Response 1- 14: Thanks for raising these questions. We have discussed this in the revised version. “Consistent with a lower migrating ability of M1 macrophages and morphometric analysis, IFN-γ treatment significantly reduced the migration of RAW264.7 and BV-2 but not 1-MT, INCB24360. The multiple filopodia of macrophage in IFN-γ treatment might contribute to the phagocytic or endocytic ability but not migrating”. 

15. The discussions need to be re written adding more references on the functional effect of the IDO-1. At the same time the shortcomings of the study and the future prospects of your study need to be mentioned.

Response 1-15: Thanks for your suggestions. Similar to the reviewer #2 views, this topic is really compelling for us. We all might have many chances to expand the topic based on our observation. In the future, to what extent IDO-1 expressing microglia/macrophages contribute to immune barriers of the brain parenchyma and if the IDO-1 expressing endothelial cells also contribute to the rejection of T cells by parenchyma? is worth exploring. Another interesting point is that if IDO-1 deregulation in macrophage/microglia correlated with brain autoimmunity diseases caused by the abnormal entrance of T cells to parenchyma, such as Multiple Sclerosis (Pilli et al., 2017).

Minor comments:

1. In abstract - Line 15: IDO-1 expression in perivascular macrophages instead of IDO-1 in macrophages. [please replace it]

Response 1-16: Thanks. We did it.

2. There are some lines in the text where IDO-1+ has been mentioned, it would be convenient to mention IDO-1

expressing cells.

Response 1-17: Thanks for pointing this out. We replaced it

3. In Fig 2 the units of the graphs need to be mentioned as fold change or log change.

Response 1-18: Thanks. The unit fold change. We added it in the revised version.

4. Abstract Line 7: “Here” - Please specify is this at the steady state? Human or mouse?

Response 1-19: Thanks. We have specified in the revised version.

5. The primer details of the house keeping gene, alpha-tubulin is missing.

Response 1-20: Thanks. Our house keeping gene in RT-PCR is GAPDH. We included the primer of “GAPDH” in previous version. Alpha-tubulin was only used for western blotting.

Reviewer #2: The current study presents intriguing finding in the brain immunology. The authors propose that IDO-1+ macrophages/microglia in meninges have higher scavenging ability which might be involved in preventing meninges/perivascular space T cells from entering the parenchyma maintaining immunosuppressive environment in the brain. While the manuscript represents an interesting finding and has a decent amount of immune fluorescence evidences and data, there are major considerations that prevents the manuscript’s publication in its current form.

Thanks for your insightful comments. Meninges immunity is intriguing for us as you, especially IDO-1+ microglia/macrophage. Dr. Lili provides us some samples of the freshly surged unaffected brain from glioma patients, making it possible to see the distribution pattern of IDO-1+ macrophages in the human brain. We are willing to move forward as quickly as possible. Unfortunately, COVID-19 changed everything, especially our lab work and students' accessibility to lab and animal models. Our work stopped for near eight months. Here we provided and shared this evidence to the scientific community to push this work forward. We know that in vivo works will provide vital pieces of evidence to IDO+ macrophages in brain immunity. We hope our data are enough to support the conclusion in this paper. 

Major Considerations:

1- The major issue in this manuscript is the lack of enough quantitative evidences to support the data. The main analysis in the study was based on manual morphometric analysis of the immune fluorescent photos. While this is a valid analysis, it is subjective and require further verification with other studies "when cells are manually selected, a heavy operator‐dependent bias is introduced, impairing procedure reproducibility (Ruffinatti, Genova et al. 2020)”.

Response #-2-1: Thanks for raising this question. We agree that on manual morphometric analysis exist some bias as other analysis. The reason we used morphometric analysis is that we found that IFN-γ treatment dramatically increased cellular size. 15-25% of IFN-γ treated cells reached 50-80µm in diameter, even though they reached 100µm in diameters. We have consulted our core facility technician. They informed us that the size of NOZZLE in core facility Flowcytometry is 100µm. The general rule of thumb is that your nozzle size should be about 4-5 times larger than the size of the cells being interrogated. Theoretically, our flow cytometry only analyzes the cell that is smaller than 33µm in diameters. If we analyze a cell with 50-80µm in diameters, we need a specialized NOZZLE. Unfortunately, our core facility is not equipped with a NOZZLE larger than 100µm. We have analyzed the dextran uptake in RAW264.7 by Flow after treatment. The trendy of data is nearly consistent to our results (Reviewer and Editor only Fig.3). Thus, we believe that our manual morphometric analysis is reliable. 

• Fig.1A, the control photo shows extensive microglia/macrophage (almost 1:2 microglia/neuron) raising a question about the brain's health tested and quality of the preparation (high background noise).

Response #-2-2: Thanks for raising this question. We have stained a brain section 50µm thick. Thick sectioned staining and Z-stack construction often visualize more processes and cells. As autofluorescent lipofuscins widely exist in the adult human brain, you need to be careful and well-trained when you read thick sections. We have sectioned some thin sections and further done H&E staining of the thin slice to confirm tissue status before our staining. If you wish to see the status of tissues, we are willing to provide them. (Reviewer and Editor only Fig.4B).

• Fig 1C, the authors need to show higher magnification of the cells without the red filter similar to the others.

Response #-2-3: Thanks for raising this question. We showed a higher magnification insert without a red filter in the revised version. 

• Fig. 1E, (Assume GBM brain, no legend for) IDO-1+ cells are negative for IBA-1. How the authors explain these findings? Although, it is reported that macrophages form about 30-50% of inflammatory infiltrate in the tumor mass and they are known to be IDO+? (Sevenich 2018)

Response #-2-4: Thanks for pointing this out. We inadvertently labeled E as F in the figure legend. We are sorry for bringing confusion. Here we corrected the inadvertent mistake in the revised version. You are right, and we observed plenty of IDO+ macrophages in pancreatic cancer, GBM, and HCC (Please refer to Editor Reviewers only figure-2). In this study, we only focused on the functions of IDO+ macrophage/microglia in physiological conditions. In the future, we will discuss the role of IDO+ macrophage in pancreatic cancer and GBM. In this part, we used the IDO+ neoplastic cells of GBM as a positive control. The academic editor also raises question about controls in immunostaining. Please refer to the reviewer and editor Fig.2-3.

Although, it is reported that macrophages form about 30-50% of inflammatory infiltrate in the tumor mass and they are known to be IDO+? (Sevenich 2018)

Thanks for raising this question. This is a big question. It is a little far from the scope of this paper. Like other tumors, brain tumors have plenty of immunes infiltrates. Based on our knowledge, the complicated network of immune cells in tumors needs more time to address by delicate and rigorous work. If we have a chance, we will do it. We wish not to include the topic of tumor in this paper [please refer to the reviewer and editor Fig.2 and Fig.3]. Thanks for understanding. 

• Fig.3A, the authors measured the control cells' filopodia. While they choose the upper cells to measure, another type of cells with extensive processes could be visualized in the same field. What is the rationale that authors used in selecting the cells for analysis?

Response #-2-5: Thanks for pointing this out. It seems that the reviewer is confused by our data. The magnified region only showed typical cellular filopodia in the macrophages in each group. The lower panel is from the upper panel, and the channel is shown as a gray color. As gray is easy to read, we changed the display pattern. The representative image does not mean that we only counted the cells; the representative image is only the image that the author thought it is typical of this group. We have counted a group of the image in our previous version. In the revised version, we have increased the counts. 

• The authors need to add the negative control for the staining to exclude nonspecific staining, for example the neurons in the same field should be negative for both staining.

Response #-2-6: This question was also raised by the academic editor. In our human brain's thick section staining system, it is difficult to do a reasonable control as in thin slide [In thin slides, you can attach two slices in one slide and used one as control]. The adult human brain neurons accumulate lipofuscins, autofluorescent particles in the neuron excited by 405, 488, and 555 lasers. However, we did not notice that large or decent lipofuscin in the microglia and astrocytes in the human brain. We have used well-tested antibodies in our research. I am a histologist who has more than 20-years’ experience in histology. We have tested our antibodies in wide range of tissues. (Reviewer and Editor only Fig.4B).

• Fig. 3A, the authors concluded on the polarization ratio between M1&M2 using morphology as a differentiating point. While there is reported difference in the morphology between M1&M2, the inverted phase morphology is not enough as the accuracy of the morphological differentiation between the two cells is based on differences in size, perimeter, shape, intensity, and texture of the actin and nuclear stain (Rostam, Reynolds et al. 2017), which wouldn’t be achieved by the inverted phase light microscope. Accordingly, other methods like phenotypic marker expression (Calprotectin, MR), cytokines analysis should be used for differentiation the two types. Also, quantitative measurement of the M1/M2 ratio using pan-macrophage markers staining and analysis by flowcytometry would have given more precise data (Antonios, Yao et al. 2013).

Response #-2-7: The living cells are versatile. The morphology of the same cells differs from lab to lab based on their culturing system and cellular density also dramatically affects cellular morphology. Moreover, classifying macrophage by its morphology is debated for many years (Martinez and Gordon, 2014). However, someone did an excellent job based on morphology (McWhorter et al., 2013). I am a histologist who has done morphological analyses for more than 20 years. We all know that the morphology of cells is still the golden standard in histology and pathology despite versatile. This study has done a good control on cellular density in the dishes and culturing system [testing infection]. Our cellular density is lower than 40% confluent in the culturing system. We used inverted microscopy to analyze the morphology, but we have applied confocal microscopy to 3D scan the cellular morphological changes after Phalloidin staining (Fig.3B, 6B). In addition, cellular migration assay data validate our morphological analysis (Please also refer to comments 2-4 or 8). As the methodology preference differs from lab to lab and group to group, we wish that the reviewer accepts our morphological analysis preference.

• Fig. 7, the authors are reporting on the dextran uptake and latex beads phagocytosis. While the photos are representative, we suggest integration. of the results with flowcytometry uptake studies.

Response #-2-8: Again. The reason we used morphometric analysis is that we found that IFN-γ treatment dramatically increased cellular size. 15-25% of IFN-γ treated cells reached 50-80µm in diameter, even though they reached 100µm in diameters. We have consulted our core facility technician. They informed us that the size of NOZZLE in Flow cytometry at core facility is 100µm. The general rule of thumb is that your nozzle size should be about 4-5 times larger than the size of the cells being interrogated. Theoretically, our flow cytometry only analyzes the cell that is smaller than 33µm in diameters. If we analyze a cell with 50-80µm in diameters, we need a specialized NOZZLE. Unfortunately, our core facility is not equipped with a NOZZLE larger than 100µm. In the revised version, we used Flow cytometry to analyze dextran uptake. The trendy of data is nearly consistent to our results (reviewer and editor only Fig.3). 

• Fig.2&5, while the cytokines’ mRNA expression is a valid representative way, including supernatant’s protein analysis using ELISA, similar to supplementary figure 7, could add to the results.

Response #-2-9: Thanks for pointing this out. Our manuscript is mainly focused on the physiological and immunological roles of IDO-1+ upregulation in microglia/macrophages. M1 and M2 are effective way to classify the physiological and immunological roles. However, this classification is controversial sometimes. Including supernatant’s protein analysis using ELISA in Fig.2&5 will strengthen our data. However, we thought that our data is enough to support our conclusion in this paper. This is a specific season, and we are willing to prepare more data for our paper. Unfortunately, you might not know the university is in a partially open condition. Thanks for your understanding.

2- The statistics needs to be revised thoroughly through the study, for example: • Fig.2B (mRNA expression level of CD206, Arg.1), there is a visible difference in the mean between control and 1-MT, yet it does not show statistical significance. This could be explained by small sample size, or unequal variability between the groups or non-normal distribution of the data. Accordingly, optimum sample size >6, blotting of the individual values and confirming the validity of the statistical test can solve this.

Response to reviewer #2-10. Thanks for pointing this out and gave us useful suggestions. We have repeated RT-PCR of CD206 and Arginase 1 (Arg1) gene and added this part to the paper. Based on your suggestions, we have modified the statistical plots of CD206 and Arg-1. We have done RT-PCR to see the transcriptional changes. As most protein levels data are consistent with the transcription changes, we thought three repeats are enough. We thought the strength of our recent data is enough to support the conclusion in this manuscript. Thanks for understanding.

 • Fig.3A, How the authors explain the variability between the sample size between groups in the same comparison? And how One way ANOVA was conducted with missing values? (control N= 7, while the IFN n=5, IMT n=4), while the legend states that n=7. This again was observed in Fig.3B, Fig.5A and Fig.6A.

Response to reviewer #2-11：

1) Thanks for raising these questions. As it is almost impossible that the cells are evenly distributed in the culturing dishes, the number of cells varied from image to image. In addition, some cells huddled together in some regions, which makes them difficult to identify and count. Thus, some figures have more countable cells than others, which causes the differences of n. We counted 500-800 cells in each group, the statistical power of 500-800 cells is enough to tell the differences. 

2) Previously, we evaluated high resolution images of over ten cells in Fig.3B. In the revised version, we added the data of another twenty cells to Fig.3B. 

• How many times the experiments were repeated?

We have repeated each experiment at least three times. After getting repeatable data, we stopped repeating.

 • At figure 6A, the authors stated that n>5 without stating what is the sample size exactly and if there is discrepancy between groups. While this is a valid way to represent sample size, however if you are going to statistically compare between them using One Way ANOVA, the exact sample size in each group should be reported. This again was observed in Fig. 4,5 and 7. 

Response to reviewer #2-12-1: Thanks for raising this question. We have repeated these experiments more than ten times. We have checked our original data, and we found we showed 4-5 times data here. Each group included 7-8 images.

To solve the previous issues, meticulous revision needs to be done to the figures and the legends. I also recommend using dot plot figure whenever possible instead of bar graph to show the individual values and adding the mean and the SEM values in the text specially with the polarization ratios.

Response to reviewer #2-12-2: Thanks for pointing these questions. We have used dot plot figures to improve the figure in the revised manuscript. In some parts, we have used bar plots. As we have several repeats in each group and protein levels and RNA levels data match with each other, we have used bar plots in some parts.

3- A previous study in 2012 has found that "IDO Expression in Brain Tumors Increases the Recruitment of Regulatory T Cells and Negatively Impacts Survival". Based on this study findings, how the authors can explain this? (Wainwright, Balyasnikova et al. 2012)

Response to reviewer #2-13: Thanks for mentioning the paper from Dr. Maciej group. We have cited this in the revised version. IDO-1 is widely expressed in human tumors, including pancreatic cancers, HCC et al., and immune infiltrates in tumors [please refer to our reviewer and editor only Figure-2-3]. It is well known that IDO-1 expression in tumors recruits regulatory T cells. We also observed that some malignant cells in GBMs are highly expressed IDO-1 consistent with Dr. Maciej group [Fig.1E]. In this work, IDO-1 expression in neoplastic cells is our positive control. The topic of IDO-1 in brain neoplastic cells is not the scope of this study. Unfortunately, most IDO-1 inhibitors in cancer trials failed (2018; Le Naour et al., 2020). In my personal view, we need to find an answer to whether IDO-1 expression neoplastic cells or immune cells in solid tumors significantly contribute to immune suppression or the creation of immune desert in solid tumors by recruiting Tregs or MDSA, including pancreatic cancers, glioma et al. We have been searching for an answer to this question for several years. You might have a chance to see our work soon (Reviewer and editor only Fig.1 and Fig.2). 

Minor Considerations:

1- Please review the references (1-4) as they do not match the text in the first paragraph. While the text mentions on IDOs, these references describe the microglia role in Parkinson's disease and the polarization of macrophage.

Response to reviewer #2-15: Thanks for pointing this out. It seems that it is a problem of the software which we used. We have freshened the citation in the revised version. 

2- In the legend of figure 1, the authors need to describe which is murine brain and which is human brain as the figures should be able to stand by itself.

Response to reviewer #2-16: Thanks. The murine brain is the mouse brain. We have specified in the revised version.

3- Figure 1E is not explained either in the results section nor the figure legend

Response to reviewer #2-17: This is an inadvertent error. This error is also noticed by reviewer #2. We mixed the labeling in the previous version and corrected it in the revised version. 

4- Fig. 2, the statistical significance was not demonstrated in some graphs.

Response to reviewer #2-18: We have added the statistics to all images in the revised version.

5- The manuscript needs to be revised for typo error like • Page 17 line 14, However However (delete one of them)

Response to reviewer #2-19: Thanks. We have corrected it in the revised version. 

• Page14 line 20, Adipogenic? (Should be Adipogen?)

Response to reviewer #2-20: Thanks. It is an inadvertent spelling mistake. We corrected it in the revised version.

References:

Antonios, J. K., Z. Yao, C. Li, A. J. Rao and S. B. Goodman (2013). "Macrophage polarization in response to wear particles in vitro." Cellular & Molecular Immunology 10(6): 471-482.

Rostam, H. M., P. M. Reynolds, M. R. Alexander, N. Gadegaard and A. M. Ghaemmaghami (2017). "Image based Machine Learning for identification of macrophage subsets." Scientific Reports 7(1): 3521.

Ruffinatti, F. A., T. Genova, F. Mussano and L. Munaron (2020). "MORPHEUS: An automated tool for unbiased and reproducible cell morphometry." Journal of Cellular Physiology 235(12): 10110-10115.

Sevenich, L. (2018). "Brain-Resident Microglia and Blood-Borne Macrophages Orchestrate Central Nervous System Inflammation in Neurodegenerative Disorders and Brain Cancer." Frontiers in Immunology 9(697).

Wainwright, D. A., I. V. Balyasnikova, A. L. Chang, A. U. Ahmed, K.-S. Moon, B. Auffinger, A. L. Tobias, Y. Han and M. S. Lesniak (2012). "IDO Expression in Brain Tumors Increases the Recruitment of Regulatory T Cells and Negatively Impacts Survival." Clinical Cancer Research 18(22): 6110.

Reviewer #3: In this manuscript Ji et al investigated differential expression of IDO1 in microglia cells in the brain parenchyma, meninges, brain tumors and brain injury models. In addition, they also investigated potential function of IDO1 in microglia/macrophages. Below are some of the concerns that need to be addressed.

1. Throughout the manuscript, please improve the writing. Logically explain why an experiment was done and what is the conclusion.

Reviewer #3-1. Thanks. We have done our best to improve in the revised version. Please refer to the revised version. Previously, the manuscript is edited by Elsevier. 

2. Rephrase the concluding paragraph in the introduction- it is confusing and hard to understand.

Reviewer #3-2. Thanks. We rewrote the conclusion in the revised version. 

3. Please explain why INF-Gamma was used- it is kind of choppy and it appears out of nowhere.

Reviewer #3-3. Thanks for pointing this out. IFN-γ is a typical cytokine that increases IDO-1 expression and activity. Microglia/macrophage are more sensitive to IFN-γ treatment. Thus, we have used it to regulate IDO-1 expression. 

4. Please include genetic inhibition of IDO1 and carry out some of the phenotypic data to provide additional evidence for IDO1 functions in microglia.

Reviewer #3-4. Thanks for pointing out this. IDO-1 is an enzyme that is a druggable target. Both 1-MT and INCB24360 are well-tested candidates to inhibit IDO-1 activity in vivo (Le Naour et al., 2020; Long et al., 2019). IDO-1 knockout mouse is commercially available now. We have checked the transcriptome data of microglia in IDO-1KO mice and found that deletion of IDO-1 does not significantly affect the expression of the typical M1 and M2 related markers (Gonzalez-Pena et al., 2016). As TDO-2 is highly expressed in the brain, it is possible that TDO-2 is possible to substitute the function of IDO-1 after deleting IDO-1. In addition, INCB and I-MT, a well-tested IDO-1 inhibitor, can inhibit both IDO-1 and TDO-2. Thus, we have used the two potential drugs to inhibit IDO-1 activity. We know that genetic manipulation is an excellent way to provide more insight into IDO-1 function in microglia. However, the strength of our data is enough to support the conclusion in this paper. 

Reference

(2018). Companies Scaling Back IDO1 Inhibitor Trials. Cancer Discov 8, OF5.

Blair, A. B., Kleponis, J., Thomas, D. L., 2nd, Muth, S. T., Murphy, A. G., Kim, V., and Zheng, L. (2019). IDO1 inhibition potentiates vaccine-induced immunity against pancreatic adenocarcinoma. J Clin Invest 129, 1742-1755.

Gonzalez-Pena, D., Nixon, S. E., Southey, B. R., Lawson, M. A., McCusker, R. H., Hernandez, A. G., Dantzer, R., Kelley, K. W., and Rodriguez-Zas, S. L. (2016). Differential Transcriptome Networks between IDO1-Knockout and Wild-Type Mice in Brain Microglia and Macrophages. PLoS One 11, e0157727.

Han, X., Ma, L., Gu, J., Wang, D., Li, J., Lou, W., Saiyin, H., and Fu, D. (2021). Basal microvilli define the metabolic capacity and lethal phenotype of pancreatic cancer. J Pathol 253, 304-314.

Hexige, S., Ardito-Abraham, C. M., Wu, Y., Wei, Y., Fang, Y., Han, X., Li, J., Zhou, P., Yi, Q., Maitra, A., et al. (2015). Identification of novel vascular projections with cellular trafficking abilities on the microvasculature of pancreatic ductal adenocarcinoma. J Pathol 236, 142-154.

Le Naour, J., Galluzzi, L., Zitvogel, L., Kroemer, G., and Vacchelli, E. (2020). Trial watch: IDO inhibitors in cancer therapy. Oncoimmunology 9, 1777625.

Long, G. V., Dummer, R., Hamid, O., Gajewski, T. F., Caglevic, C., Dalle, S., Arance, A., Carlino, M. S., Grob, J. J., Kim, T. M., et al. (2019). Epacadostat plus pembrolizumab versus placebo plus pembrolizumab in patients with unresectable or metastatic melanoma (ECHO-301/KEYNOTE-252): a phase 3, randomised, double-blind study. Lancet Oncol 20, 1083-1097.

Ma, L., Hu, B., Liu, Y., Vermilyea, S. C., Liu, H., Gao, L., Sun, Y., Zhang, X., and Zhang, S. C. (2012). Human embryonic stem cell-derived GABA neurons correct locomotion deficits in quinolinic acid-lesioned mice. Cell Stem Cell 10, 455-464.

Martinez, F. O., and Gordon, S. (2014). The M1 and M2 paradigm of macrophage activation: time for reassessment. F1000Prime Rep 6, 13.

McWhorter, F. Y., Wang, T., Nguyen, P., Chung, T., and Liu, W. F. (2013). Modulation of macrophage phenotype by cell shape. Proc Natl Acad Sci U S A 110, 17253-17258.

Pilli, D., Zou, A., Tea, F., Dale, R. C., and Brilot, F. (2017). Expanding Role of T Cells in Human Autoimmune Diseases of the Central Nervous System. Front Immunol 8, 652.

Editor and Reviewer only Figure-1 

Fig.1 IDO1 and TDO were highly expressed in human PDAC tissue.

A, IDO1 expression pattern in precancerous pancreatic tissues, primary tumor and invasive lesions. 

B. IDO-1 expression in extrusion cells.

Editor and Reviewer only Figure-2. IDO-1 is highly expressed in M1-like macrophage in the invasive region or metastatic site of pancreatic cancers

A. The representative images of E-cadherin, IDO-1, and CD45RA staining in pancreatic cancer tissues and the lymph nodules (I and II, a lymph nodule with metastatic cells; III, invasive duct; IV, precursor lesion; yellow arrows, immune cells; white arrows, neoplastic cells). N, 10. 

B. Comparing counts of immune cells with IDO-1 higher in lymph nodules with metastatic neoplastic cells with that in lymph nodules without metastatic neoplastic cells 

C. Calibrating the size of IDO-1 high and CD45RA weak or IDO-1 low and CD45RA high cells (the longest diameter of cells). Student t-test.

E. The representative images of CD11B and CD45RA staining in pancreatic cancer tissues and the lymph nodule (Inner insert, typical macrophage; yellow arrows, macrophage with large body size and lower CD45RA levels; white arrow macrophage with a smaller body size and higher CD45RA levels). Patient number, 6. 

F. Calibrating the size of CD11B high and CD45RA weak or CD11B high and CD45RA high cells (the longest diameter of cells). Student t-test.

G. Representative image of CD16 and IDO-1 staining in pancreatic cancer tissues (I, surrounded a neoplastic duct; II, an inflammatory site). N, 6.

Reviewers and Editor Only Figure -3

A. Dextran uptake in RAW264.7 detected by Flow cytometry.

Reviewer and editor only figure 4. 

A. The immunofluorescent staining control in mouse (omitting primary antibody, Donkey anti-goat Alexa 594; Donkey anti-rabbit Alexa 488).

B. IDO-1 and IBA-1 antibody immunostaining in human brain cortex (Yellow arrow, neuron).

---

## [Decision Letter · Decision Letter 1]

9 Aug 2021

PONE-D-21-06075R1

Characterization of IDO-1 expressing macrophages/microglia in the meninges and perivascular space of the human and murine brain

PLOS ONE

Dear Dr. Saiyin,

Thank you for submitting your manuscript to PLOS ONE. After careful consideration, we feel that it has merit but does not fully meet PLOS ONE’s publication criteria as it currently stands. Therefore, we invite you to submit a revised version of the manuscript that addresses the points raised during the review process.

Considering the nature of the data presented in your manuscript, in our assessment, the title of this manuscript is too broad and unsupported by the evidence, and hence can be misleading. Only Figure. 1 has data supporting of expression of IDO-1 in the meninges and perivascular space of the human and murine brain. Rest of the data in the paper describe the effect of inhibiting IDO-1 in RAW264.7/BV-2 cells. While these cells can offer good system to study effect of IDO-1 inhibition, the findings from these cells cannot be meaningfully extrapolated to microglia in vivo.<o:p></o:p>

Therefore, we ask that you modify title such that the presented evidence in the manuscript adequately supports it. In modifying your title, we specifically ask that the RAW264.7/BV-2 cells be identified in the title itself. The current title warrants additional in vivo studies examining effect of IDO-1 inhibition and its physiological implications.  We look forward to receiving your revised manuscript.

Kind regards,

Nagaraj Kerur

Academic Editor

PLOS ONE

Reviewers' comments:

Reviewer's Responses to Questions

**Comments to the Author**

1. If the authors have adequately addressed your comments raised in a previous round of review and you feel that this manuscript is now acceptable for publication, you may indicate that here to bypass the “Comments to the Author” section, enter your conflict of interest statement in the “Confidential to Editor” section, and submit your "Accept" recommendation.

Reviewer #1: All comments have been addressed

Reviewer #2: All comments have been addressed

Reviewer #3: All comments have been addressed

2. Is the manuscript technically sound, and do the data support the conclusions?

Reviewer #1: Yes

Reviewer #2: Yes

Reviewer #3: Yes

3. Has the statistical analysis been performed appropriately and rigorously? 

Reviewer #1: Yes

Reviewer #2: Yes

Reviewer #3: Yes

4. Have the authors made all data underlying the findings in their manuscript fully available?

Reviewer #1: Yes

Reviewer #2: Yes

Reviewer #3: Yes

5. Is the manuscript presented in an intelligible fashion and written in standard English?

Reviewer #1: Yes

Reviewer #2: Yes

Reviewer #3: Yes

6. Review Comments to the Author

Reviewer #1: The authors have made all changes necessitated by the first round of review process. All additional data presented by the authors fully support the authors conclusion. I have no further comments or concerns.

Reviewer #2: The authors addressedl the main concerns. Although this paper could be supplemented with stronger evidences, the theory presented opens new scopes in this field for further explorations.

Reviewer #3: The authors have done a good job of addressing the questions raised by the reviewers. However, I felt the authors were adamant and stubborn in their responses. I wish they could use their wisdom to respond the questions raised by the reviewers.

7. PLOS authors have the option to publish the peer review history of their article (what does this mean?). If published, this will include your full peer review and any attached files.

Reviewer #1: No

Reviewer #2: No

Reviewer #3: **Yes: **Mallikarjun Patil

---

## [Author Response · Author response to Decision Letter 1]

10 Aug 2021

Response to editor and reviewers

Thank you for reviewing and revising our manuscript. Based on your suggestions comments, we have made the revision again. We all hope that our re-revised version fully addresses your and reviewers’ concerns and meets PLOS ONE’s publication criteria. If you have any further concern, we are willing to address as soon as possible. 

Editor comments and response to the concerns:

Thank you for submitting your manuscript to PLOS ONE. After careful consideration, we feel that it has merit but does not fully meet PLOS ONE’s publication criteria as it currently stands. Therefore, we invite you to submit a revised version of the manuscript that addresses the points raised during the review process.

Considering the nature of the data presented in your manuscript, in our assessment, the title of this manuscript is too broad and unsupported by the evidence, and hence can be misleading. Only Figure. 1 has data supporting of expression of IDO-1 in the meninges and perivascular space of the human and murine brain. Rest of the data in the paper describe the effect of inhibiting IDO-1 in RAW264.7/BV-2 cells. While these cells can offer good system to study effect of IDO-1 inhibition, the findings from these cells cannot be meaningfully extrapolated to microglia in vivo.

Therefore, we ask that you modify title such that the presented evidence in the manuscript adequately supports it. In modifying your title, we specifically ask that the RAW264.7/BV-2 cells be identified in the title itself. The current title warrants additional in vivo studies examining effect of IDO-1 inhibition and its physiological implications. We look forward to receiving your revised manuscript.

Response to Dr. Kerur:

Thank you for taking the time to review our manuscript rigorously. We all greatly appreciate your constructive suggestions and comments. We have re-revised our manuscript based on your and reviewer comments. We all wish our revision fully meets PLOS ONE’s publication criteria.

We totally agree with your suggestion and comment that including BV-2 and RAW264.7 in the manuscript title. As you mentioned, our data in this manuscript did not fully support our previous title. The new title is “Characterizing the distributions of IDO-1 expressing macrophages/microglia in human and murine brain and evaluating the immunological and physiological roles of IDO-1 in RAW264.7/BV-2 cells”. Our title is flexible. If you have any good suggestion about our title, we are willing to accept or modify based on your suggestion.

Reviewers' comments:

Reviewer's Responses to Questions

Comments to the Author

1. If the authors have adequately addressed your comments raised in a previous round of review and you feel that this manuscript is now acceptable for publication, you may indicate that here to bypass the “Comments to the Author” section, enter your conflict of interest statement in the “Confidential to Editor” section, and submit your "Accept" recommendation.

Reviewer #1: All comments have been addressed

Thanks again.

Reviewer #2: All comments have been addressed

Thanks again.

Reviewer #3: All comments have been addressed

Thanks again.

2. Is the manuscript technically sound, and do the data support the conclusions?

Reviewer #1: Yes

Reviewer #2: Yes

Reviewer #3: Yes

3. Has the statistical analysis been performed appropriately and rigorously?

Reviewer #1: Yes

Reviewer #2: Yes

Reviewer #3: Yes

4. Have the authors made all data underlying the findings in their manuscript fully available?

Reviewer #1: Yes

Reviewer #2: Yes

Reviewer #3: Yes

5. Is the manuscript presented in an intelligible fashion and written in standard English?

Reviewer #1: Yes

Reviewer #2: Yes

Reviewer #3: Yes

6. Review Comments to the Author

Reviewer #1: The authors have made all changes necessitated by the first round of review process. All additional data presented by the authors fully support the authors conclusion. I have no further comments or concerns.

Dear reviewer #1: Thanks again. Your comments are helpful to better our manuscript. 

Reviewer #2: The authors addressed the main concerns. Although this paper could be supplemented with stronger evidences, the theory presented opens new scopes in this field for further explorations.

Dear reviewer #2: Thanks for the insightful comments about our manuscript. Your concerns about this paper are also our primary concern in the future. After this pandemic is over, we will address the immunological role of IDO+ macrophage/microglia in meninges and perivascular space, even the immune homeostasis in the parenchyma in our subsequent work. We are preparing more in vivo experiments to address your and our concerns about IDO-1+ macrophages/microglia in the brain. 

Reviewer #3: The authors have done a good job of addressing the questions raised by the reviewers. However, I felt the authors were adamant and stubborn in their responses. I wish they could use their wisdom to respond the questions raised by the reviewers.

Dear Dr. Mallikarjun Patil:

Thanks for taking the time to review our manuscript. In this specific season, we try our best to revise our manuscript. The availability of core facilities, especially FACS, is limited because of some restrictions and regulations. As non-native speakers of English, it is a little difficult for us to fluently and smartly communicate with reviewers or editors as a native speakers in the scientific community. In the future, we will try our best to improve our communication skills in the scientific literature, and you will have a chance to a better work about meningeal immunity soon. If some responses are inappropriate, it might be caused by our communication skills but not our true intention. Thanks for raising these questions. We will try our best to respond to the comments of reviewer and editor with wisdom in the future.

---

## [Editor Report · Decision Letter 2]

22 Sep 2021

Characterizing the distributions of IDO-1 expressing macrophages/microglia in human and murine brains and evaluating the immunological and physiological roles of IDO-1 in RAW264.7/BV-2 cells

PONE-D-21-06075R2

Dear Dr. Saiyin

We’re pleased to inform you that your manuscript has been judged scientifically suitable for publication and will be formally accepted for publication once it meets all outstanding technical requirements.

Kind regards,

Nagaraj Kerur

Academic Editor

PLOS ONE
---

## [Editor Report · Acceptance letter]

27 Oct 2021

PONE-D-21-06075R2 

Characterizing the distributions of IDO-1 expressing macrophages/microglia in human and murine brains and evaluating the immunological and physiological roles of IDO-1 in RAW264.7/BV-2 cells 

Dear Dr. Saiyin:

I'm pleased to inform you that your manuscript has been deemed suitable for publication in PLOS ONE. Congratulations! Your manuscript is now with our production department. 

Kind regards, 

on behalf of

Dr. Nagaraj Kerur 

Academic Editor

PLOS ONE